# Deconvoluting Cr states in Cr-doped $UO_2$ nuclear fuels via bulk and single crystal spectroscopic studies

Gabriel L. Murphy [1] ✉, Robert Gericke[2], Sara Gilson[2], Elena F. Bazarkina[2,3], André Rossberg[2,3], Peter Kaden [2], Robert Thümmler[1], Martina Klinkenberg[1], Maximilian Henkes[1], Philip Kegler[1], Volodymyr Svitlyk [2,3], Julien Marquardt [4], Theresa Lender [5], Christoph Hennig [2,3], Kristina O. Kvashnina [2,3] & Nina Huittinen [2] ✉

Cr-doped $UO_2$ is a leading accident tolerant nuclear fuel where the complexity of Cr chemical states in the bulk material has prevented acquisition of an unequivocal understanding of the redox chemistry and mechanism for incorporation of Cr in the $UO_2$ matrix. To resolve this, we have used electron paramagnetic resonance, high energy resolution fluorescence detection X-ray absorption near energy structure and extended X-ray absorption fine structure spectroscopic measurements to examine Cr-doped $UO_2$ single crystal grains and bulk material. Ambient condition measurements of the single crystal grains, which have been mechanically extracted from bulk material, indicated Cr is incorporated substitutionally for $U^{+4}$ in the fluorite lattice as $Cr^{+3}$ with formation of additional oxygen vacancies. Bulk material measurements reveal the complexity of Cr states, where metallic Cr ($Cr^0$) and oxide related $Cr^{+2}$ and $Cr^{+3}_2O_3$ were identified and attributed to grain boundary species and precipitates, with concurrent $(Cr^{+3}_xU^{+4}_{1-x})O_{2-0.5x}$ lattice matrix incorporation. The deconvolution of chemical states via crystal vs. powder measurements enables the understanding of discrepancies in literature whilst providing valuable direction for safe continued use of Cr-doped $UO_2$ fuels for nuclear energy generation.

Additive-based $UO_2$ nuclear fuels, which incorporate small amounts of metal oxides such as chromia ($Cr_2O_3$) are an important class of accident tolerant fuels (ATFs) used currently and expected to have extensive utilisation within next-generation power reactor fleets[1,2]. The introduction of additives like $Cr_2O_3$, into $UO_2$ during fuel fabrication results in increased grain growth resulting in enhanced properties. These include enhanced fission gas retention, increased plasticity and improved fuel-cladding interactions leading to overall improved in-reactor fuel efficiency over conventional $UO_2$ in power reactor applications[2–4]. Accordingly, Cr-doped $UO_2$ fuels have garnered strong interest from industry[5] and researchers alike, where experimental and theoretical studies have focused on such topics as preparation[6,7], structure[8,9], microstructure[10,11], stability[12,13], diffusion dynamics[14], and thermodynamics[15].

The determination of the Cr oxidation state within the $UO_2$ lattice matrix and its mechanism for incorporation has drawn

[1]Institute of Energy and Climate Research (IEK-6), Forschungszentrum Jülich GmbH, 52428 Jülich, Germany. [2]Institute of Resource Ecology, Helmholtz-Zentrum Dresden-Rossendorf, 01328 Dresden, Germany. [3]The Rossendorf Beamline at ESRF, The European Synchrotron, CS40220, 38043, Grenoble Cedex 9, France. [4]Institut für Geowissenschaften, Goethe-Universität Frankfurt, 60438 Frankfurt am Main, Germany. [5]Institut für Kristallographie, RWTH Aachen University, 52066 Aachen, Germany. ✉e-mail: g.murphy@fz-juelich.de; n.huittinen@hzdr.de

considerable interest and debate. The stability of its electronic configuration in its trivalent oxidation state, [Ar]3d[3], which is the form used as an additive in industrial fuel fabrication[1] and commonly in model system studies, has contributed to many bulk analysis-based investigations to conclude it exists as $Cr^{+3}$ within the $UO_2$ matrix[15–18]. Contemporary investigations have challenged this position where the utilisation of simulation methods have proposed more reduced states of $Cr^{+2}$ and $Cr^{+1}$ within the matrix[8,19]. The lack of unequivocal understanding of the redox chemistry and mechanism for Cr incorporation into $UO_2$ can be traced to its limited solubility and the complexity of chemical states it can adopt. The small size of the Cr cation compared to U leads to a low solubility that is highly dependent on the oxygen potential ($\mu_{O2}$), temperature and synthesis method[3,6,10,15,20]. It has been established that between 1500–2000 °C and $-460 < \mu_{O2} < -360$ kJ/mol, solubilities can range between 650 and 1020 ppm as Cr[3,15]. In the case of 1700 °C and $\mu_{O2}$ of $-420$ kJ/mol, it is understood to result in a Cr solubility of ~750 ppm in $UO_2$[15], where this can be typically increased via higher temperatures or higher $\mu_{O2}$. The incorporation process is further complicated during fuel sintering, where at high temperatures a variety of Cr states can be accessed, including eutectic compositions, (liquid CrO) metallic Cr, and $Cr_2O_3$, where the proportionality of these is highly dependent upon specific temperature and $\mu_{O2}$[6,10,21,22]. Indeed, the existence of dissolved Cr in $UO_2$ by means of its equilibrium with either $Cr_2O_3$ or liquid CrO at high temperatures is understood to contribute to the enhanced grain growth phenomena, which promotes the improved properties of Cr-doped $UO_2$ as a nuclear fuel[10,15]. Upon cooling, these variable Cr chemical states with different redox behaviour report to a variety of locations within the fuel structure, including as precipitates, grain boundaries and within the fuel matrix lattice[3,10,15]. Subsequently, bulk Cr-doped $UO_2$ can be considered a complex system with respect to the variable chemical states Cr can simultaneously adopt. This creates a considerable experimental challenges in redox state determination in both fresh and irradiated Cr-doped $UO_2$ nuclear fuel using bulk material measurements, which most previous investigations have used, as it becomes difficult to reliably deconvolute specific Cr chemical states and environments from others within bulk $UO_2$.

Reliably and incontrovertibly determining the chemical states of Cr in $UO_2$ requires experimental techniques which have the resolution and deconvolution ability to focus on individual chemical environments of Cr, free of parasitic interfering signals from others. Mieszczynski et al.[16] attempted this via microprobe X-ray absorption spectroscopic measurements (µ-XAS) to examine grains within irradiated and non-irradiated Cr-doped $UO_2$. Although similar results were found between fresh and irradiated Cr-doped $UO_2$, they identified significant parasitic interfering signals from Cr precipitates and grain boundary phases acting along the nanometre scale. This prevented the accurate and precisely resolved measurement of the Cr chemical state within the fuel matrix. Their work emphasised the complexity of Cr chemical states within Cr-doped $UO_2$ and particularly the challenge of deconvoluting them via bulk measurements.

In the present investigation, we provide a solution to the convoluted bulk measurement dilemma by spectroscopically examining at ambient conditions mechanically extracted single crystal grains from bulk material (sintered pellets) which are free of additional secondary Cr bearing phases and related impurities, and thus representative of the lattice matrix. The spectroscopic techniques include electron paramagnetic resonance (EPR), high-energy resolution fluorescence detection X-ray absorption near edge structure (HERFD-XANES)[23] and extended X-ray absorption fine structure (EXAFS). We demonstrate that by extracting representative single crystal grains from the bulk material, Cr chemical states can be deconvoluted, precisely identified and determined.

## Results

### Synthesis and materials characterisation

Bulk Cr-doped $UO_2$ was synthesised using an established method[6] involving coprecipitation with the addition of 3500 ppm as $Cr_2O_3$ (2395 ppm as Cr elemental). Sintering conditions of 1700 °C with a controlled $\mu_{O2}$ of $-420$ kJ/mol were used, which according to literature resulted in an approximate targeted 750 ppm solubility of Cr as elemental[15]. Supersaturated conditions were chosen in order to maximise crystal grain measurement resolution despite the loss of Cr from potential volatilisation[24]. Synchrotron X-ray powder diffraction indicated the formation of single-phase material (Supplementary Information Note 1). The extraction of the single crystal grains from sintered pellets was achieved directly via mechanical separation (described in the "Methods" section) and accordingly is completely representative of the bulk material. The single crystal grains were examined via single crystal X-ray diffraction and scanning electron microscopy (SEM) with energy dispersive X-ray spectroscopy measurements (EDS) to confirm composition and structure (Supplementary Information Notes 3 and 2) and electron paramagnetic resonance (EPR) spectroscopy to confirm the absence of secondary Cr phases (discussed in detail later). These measurements indicated that the single crystal grains were pure single crystal $UO_2$ fluorite structured in space group $Fm\bar{3}m$. No parasitic Cr near surface precipitate phases such as $Cr_2O_3$ were detected by EDS or SC-XRD and possible subsurface precipitates were further ruled out by EPR. These results were further supported by high-energy resolution fluorescence detection X-ray spectroscopy (HERFD-XAS) measurements (discussed later), indicating the acquisition of Cr-doped $UO_2$ single crystal grains free of secondary Cr phases. Parasite phases were readily identified in the bulk material as illustrated in Fig. 1, which compares and illustrates an extracted single crystal grain against the bulk parent material from SEM-EDS analysis, highlighting the absence of these parasitic phases in the extracted crystal specimen. The occurrence of these parasitic phases which can be observed in grain boundaries, as precipitates and within voids, is consistent with previous literature[6,10] regarding the distribution of secondary Cr phases in the bulk material.

### Electron paramagnetic resonance (EPR) spectroscopy

The single crystal grains, mechanically extracted from bulk Cr-doped $UO_2$ pellets were examined with EPR. EPR is a spectroscopic technique that is sensitive to unpaired electron systems, where it measures the energy transition between electron spin energy levels using variable magnetic fields. Pertinent and relevant to Cr-doped $UO_2$, it is highly sensitive to the occurrence of $Cr^{+3}$ and $Cr^{+5}$[25–27] but also to $U^{+5}$, even at trace ppm levels[28]. As shown in Fig. 2a, the EPR spectrum along with the fitted model of a Cr-doped $UO_2$ single crystal grain results in a broad axial signal of $g_{\parallel} = 3.70$ and $g_{\perp} = 2.40$. These values and line shapes with respect to literature are consistent with $Cr^{+3}$ being in a highly distorted substitutional octahedral environment[26, 27]. From EPR analysis, the overall spin could be fitted with $S = 3/2$. This can be interpreted as a high-spin $Cr^{+3}$ together with oxygen vacancy defects in the vicinity of the substituted Cr site to balance the charge[26,27]. $Cr^{+3}$ coupled with oxidised uranium such as $U^{+5}$, seen in $Ln^{+3}$ doping of $UO_2$[29] ($Ln^{+3}$ = trivalent lanthanide cation), for charge compensation, would result in an $S = 2$ spin-state (or $S = 1$, due to spin pairing) and can be excluded, as this would lead to a silent EPR spectrum. Uncoupled $Cr^{+3}$ with $U^{+5}$, would result in a second sharp signal found at lower $g$-value[28]. As only a single broad signal is found in the EPR spectra of Fig. 2a, the occurrence of $U^{+5}$ in Cr-doped $UO_2$ is excluded and can only be attributed to $Cr^{+3}$ and a local oxygen vacancy. Moreover, the formation of the oxygen vacancy cannot involve the formation of paramagnetic defects, such as +1 charged $(V_O^{\bullet})$ vacancies with unpaired electrons, as they would be readily observable by EPR, producing a sharp signal in the EPR spectrum at ~$g = 2$[30]. This implies that the generated oxygen vacancies following $Cr^{+3}$ substitution in the $UO_2$ in the post-sintered

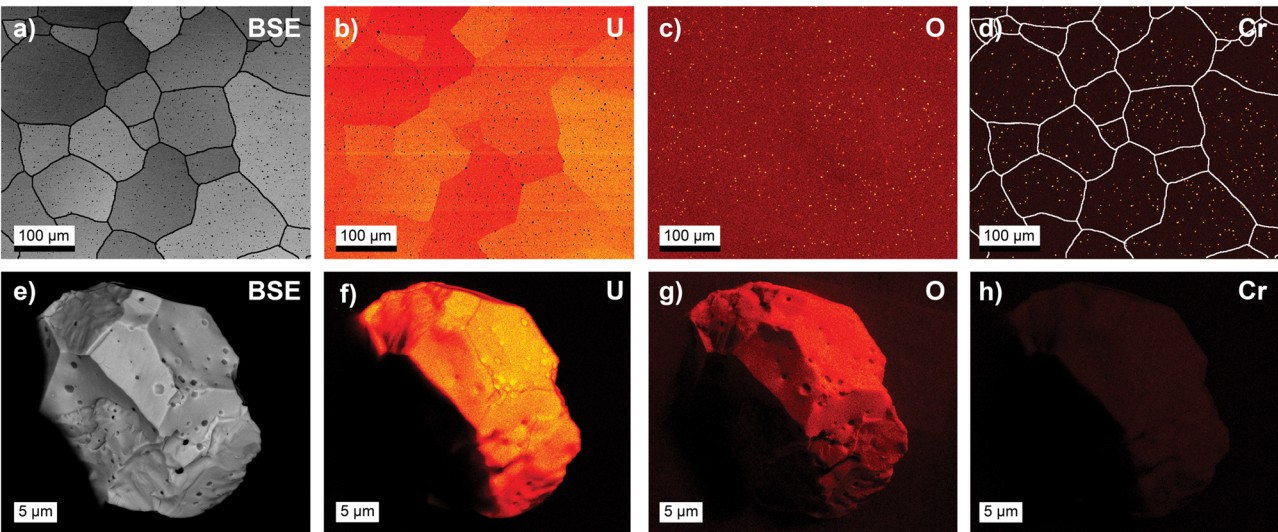

**Fig. 1 | Electron micrographs of bulk Cr-doped UO₂ and a single crystal extracted from the parent material.** BSE (**a**, **e**) and EDS (**b**–**d**, **f**–**h**) U-L, O-K and Cr-K edge images of Cr-doped UO₂ pellet (top) with 3500 ppm addition as Cr₂O₃ shown comparatively with a single crystal grain (bottom) extracted from the same material. The grain boundary map of the BSE Cr-doped UO₂ pellet has been superimposed on the Cr-K edge EDS map to highlight the distribution of Cr–O phases. Note the lack of impurity grains and Cr additions on the surface of the single crystal grain, particularly in void regions, compared to the pellet where considerable Cr oxide and metallic are present in grain boundary regions and as precipitates. The dark spots in the single crystal grain BSE are due to voids and note the shadow effect from EDS present in the U-L, O-K and Cr-K edges.

measured material lattice must be +2 charged ($V_O^{\cdot\cdot}$) or neutrally charged ($V_O^x$) on average. As vacancies are highly mobile in contrast to the $Cr^{+3}$ cations, particularly during sintering[31], the resulting charge in the lattice is globally balanced, although local charge imbalances may exist[32]. The occurrence of neighbouring twin $Cr^{+3}$ cations with a neutral oxygen vacancy is further ruled out as the associated detectable lifted anti-ferromagnetic coupling[25] would result in a considerably different spectrum with a lower *g*-value (see Supplementary Information Note 4). Hyperfine spectra could be possible due to electron coupling to the isolated $Cr^{+3}$ centres, however, it is not observed in the EPR spectra of the Cr-doped UO₂ single crystals under measurement conditions. Surface and subsurface contamination, such as from Cr₂O₃ can be excluded further from being present within or attached to the crystals as their occurrence would result in additional signals in the spectra that have different line shapes and are at lower *g* values (-1.9–2.4) (see Supplementary Information Note 4)[27]. If only $Cr^{+2}$ were to be occurring in the UO₂ lattice, this would result in an EPR silent spectrum, which is inconsistent with Fig. 2a. Consequently, EPR analysis indicates that Cr incorporation into the UO₂ lattice matrix can be described as $(Cr^{+3}_xU^{+4}_{1-x})O_{2-0.5x}$, in which the occurrence of charge balancing $U^{+5}$, lone $Cr^{+2}$ or locally twin clustered $Cr^{+3}$ centres with a neutral vacancy are all not supported.

## High energy resolution fluorescence detection X-ray absorption near-edge (HERFD-XANES) spectroscopy

To further support the mechanism for incorporation of Cr into the UO₂ matrix provided by EPR measurements, single crystal grains and bulk powder material were examined via HERFD-XANES measurements. Conventional XANES measurements of transition metal elements contain significant structural details, this includes information about both the geometric coordination symmetry and oxidation state of the element of interest. X-ray spectroscopy is frequently used to assign oxidation states in transition metal compounds, due to its element selectivity and core electron excitations, which together provide the experimental evidence of the oxidation state change. The metal K-edge (like the Cr K-edge) is often used as a fingerprint to track changes in the oxidation state via the dipole-allowed $1s–4p$ transitions that shift by several eV per oxidation state. Such a shift of the main

absorption peak reflects an increased electron affinity of the metal in the core-excited states compared to the ground state. However, extracting information with conventional XANES for transition metals can be difficult due to the core–hole lifetime broadening of the ground state which smears and reduces the measurement resolution[33]. Furthermore, the background produced by other elements can smear the recorded features as well[33]. HERFD-XANES overcomes this issue by using the intensity of the emitted X-ray fluorescence in a narrow energy bandwidth, increasing measurement resolution[33].

We provide HERFD-XANES measurements at the Cr K-edge for several Cr references and for the Cr-doped UO₂ system, including the single crystal grain and bulk material. Prior to the present investigation, all published literature on Cr-doped UO₂ was performed using conventional XANES for Cr. Accordingly, Cr K-edge HERFD-XANES measurements are provided in Fig. 2b for Cr-doped UO₂, bulk powder material and a single crystal grain with the standards: metallic $Cr^0$, $Cr^{+2}Cl_2$, $Cr^{+3}Cl_3\cdot6H_2O$, and $Cr^{+3}_2O_3$, from 5985 to 6060 eV using a step size of 0.5 eV. It should be noted, that CrCl₂ is extremely sensitive to oxidation. To prevent potential oxidation of the standard, it was prepared using an Ar-filled glovebox containing <1 ppm of O₂/H₂O with double-layer confinement to prevent air intrusion. Thereby, Fig. 2b shows the Cr K-edge HERFD-XANES spectrum recorded on CrCl₂ in its pure +2 oxidation state. We observed the strongest peak at 5989 eV whereas the second strongest is found at 5987 eV for CrCl₂. The Cr₂O₃ spectrum shows two pre-edge peaks at -5990.5 and -5987.5 eV. For $Cr^{+3}Cl_3\cdot6H_2O$, it shows a pre-edge peak at -5988 eV. The small energy difference recorded at the pre-edge between the two references in +2 and +3 oxidation states indicates that pre-edge region potentially contains fingerprint information on the oxidation state changes of complex Cr systems, like bulk Cr-doped UO₂. Pertinently, other factors like the presence of various ligands, coordination number, and metal spin state will contribute to the changes of the pre-edge features and must be examined and considered in depth before the generalised measure of oxidation state is applied to the pre-edge.

Inspecting Fig. 2b it can be observed that there is a clear shift of the main edge transitions between metallic Cr ($Cr^0$), $Cr^{+2}Cl_2$, $Cr^{+3}Cl_3\cdot6H_2O$ and $Cr^{+3}_2O_3$. The first metallic transition appears at 5991 eV, while the CrCl₂ spectrum shows two main peaks at 5993 eV

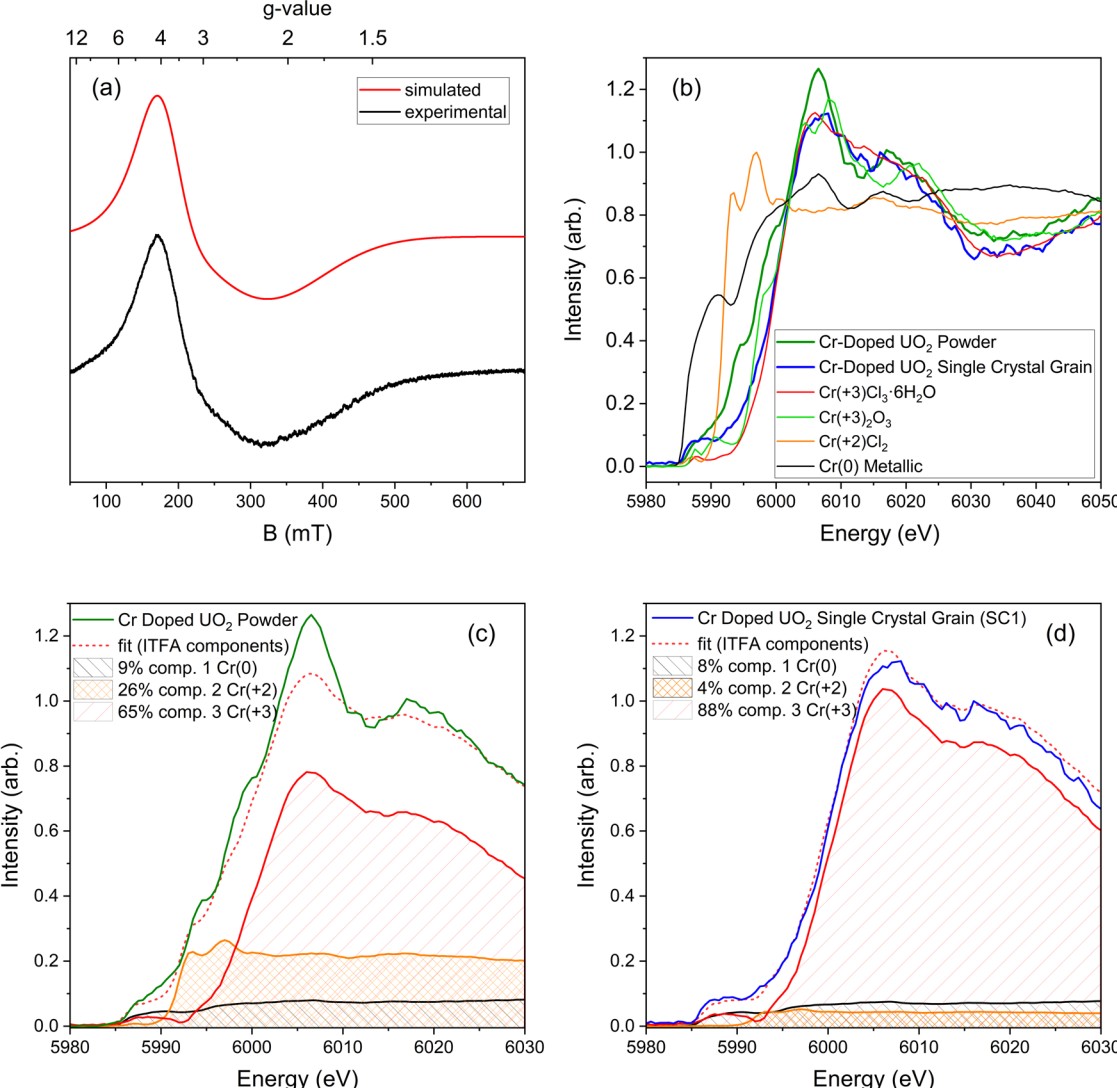

**Fig. 2 | EPR and HERFD-XANES spectroscopic analysis of Cr-doped UO₂ single crystals and bulk material. a** Measured and modelled EPR spectra of Cr-doped UO₂ single crystal grains at room temperature. **b** Normalised Cr K-edge HERFD-XANES spectra in the range 5980–6050 eV for Cr-doped UO₂ as powder and single crystal grain, $Cr^0$ (metallic), $Cr^{+2}Cl_2$, $Cr^{+3}Cl_3·6H_2O$, and $Cr^{+3}_2O_3$. **c** Normalised Cr K-edge HERFD-XANES spectra in the range 5980–6030 eV for Cr-doped UO₂ bulk powder with calculated complementary iterative transformation factor analysis (ITFA) components for $Cr^0$ (metallic), $Cr^{+2}Cl_2$ and $Cr^{+3}Cl_3·6H_2O$ and **d** Normalised Cr K-edge HERFD-XANES spectra in the range 5980–6030 eV for Cr-doped UO₂ single crystal grain with calculated complementary ITFA components for $Cr^0$ (metallic), $Cr^{+2}Cl_2$ and $Cr^{+3}Cl_3·6H_2O$.

and 5997 eV. The main edge excitations shift further to the highest incident energy for the $Cr^{+3}$ compounds. Cr K-edge main peak of $CrCl_3$ is at 6006 eV, while the K-edge spectrum recorded on $Cr_2O_3$ shows a splitting of the main edge features to transitions at ~6004.5 and ~6008.5 eV. These two transitions reflect the crystal field splitting of the $4p$ states in $Cr_2O_3$ local coordination. At the same time, the position of the main edge transitions appears at the same energy range for $Cr^{+3}Cl_3·6H_2O$ and $Cr_2O_3$, indicating that Cr K-edge HERFD-XANES can be used as a fingerprint for the oxidation state determination of complex Cr-containing systems.

Comparing the Cr-doped UO₂ single crystal and bulk powder systems HERFD-XANES spectra with that of the Cr references, provided in Fig. 2b, it can be observed that there are significant differences between the spectra of the Cr-doped UO₂ single crystal grain and the powder. It should be noted that the variation in the signal-to-noise ratio of the Cr-doped UO₂ single crystal grain line shape compared to the powder is a consequence of the much lower measurable amount of Cr contained in the ~40 μm crystal compared to the milligram quantity bearing powder. The Cr-doped UO₂ single crystal grain spectrum follows closely that of the standard $Cr^{+3}Cl_3·6H_2O$ throughout the main and post-edge region (in the energy range of 6006–6008 eV), strongly indicating the crystal contains $Cr^{+3}$, consistent with EPR analysis. In contrast, the Cr-doped UO₂ powder is observed to shift to lower energy compared to the single crystal grain. The spectrum recorded on Cr-doped UO₂ powder contains 2 distinct features at the ~5993 and ~6000 eV, which match with the main absorption features of the $CrCl_2$ reference. Furthermore, the line shape of Cr-doped UO₂ powder is less consistent with the $Cr^{+3}Cl_3·6H_2O$ standard and appears to contain features resembling those of $Cr^0$, $Cr^{+2}$ and $Cr^{+3}$, which, considering the SEM-EDS analysis (Fig. 1) and also literature[6, 10,21], is attributed to metallic and oxide precipitates and grain boundary phases. The spectrum recorded on the Cr-doped UO₂ single crystal grain exhibits a prominent peak at ~5991 eV that is also present in the metallic Cr. However, the peaks at the 5993 and 5997 eV that belong to $CrCl_2$ are absent in the Cr-doped UO₂ single crystals spectrum. This indicates and further supports the premise that Cr in the bulk powder differs considerably compared to the single crystal grain in terms of the variety of Cr oxidation and chemical states present.

In order to determine the exact contribution of the different species to the spectra of the Cr-doped $UO_2$ powder and single crystal grain, iterative transformation factor analysis (ITFA)[34] was performed and results were provided in Fig. 2c and d, respectively. The analysis shows that Cr-doped $UO_2$ powder contains 9% metallic Cr ($Cr^0$), 26% $Cr^{+2}$ and 65% $Cr^{+3}$. Notably, ITFA cannot discern the $Cr^{+3}$ contributions arising from $Cr^{+3}_2O_3$ and $(Cr^{+3}_xU^{+4}_{1-x})O_{2-0.5x}$, where 65% is a summation of their total contributions.[34] Nevertheless, when the results of the ITFA for the Cr-doped $UO_2$ bulk material are compared against the approximate solubility of Cr into $UO_2$ provided by previous experimental and thermodynamic estimates[3,15], accounting for the synthesis conditions used (1700 °C, −420 kJ/mol, 3500 ppm Cr as $Cr_2O_3$/ 2395 ppm as Cr elemental), including volatilisation which can be as high as 30%[24], and that the samples were cooled under gas atmosphere, they are strikingly consistent with the quantitative distribution of Cr phases[3,15]. From the conditions used in the present study, a 30% loss of Cr via volatilisation would result in 1676 ppm as Cr in the bulk material, in which the ITFA analysis implies 1089 ppm Cr as $Cr^{+3}$ (65%) that would distribute between the $Cr^{+3}_2O_3$ and $(Cr^{+3}_xU^{+4}_{1-x})O_{2-0.5x}$ phases. Considering a solubility limit of 750 ppm of Cr into the $UO_2$ lattice provided by the thermodynamic model of Riglet-Martial et al.[15] relevant to the conditions used in this investigation, a distribution of $(Cr^{+3}_xU^{+4}_{1-x})O_{2-0.5x}$, $Cr^{+3}_2O_3$, $Cr^{+2}$ and $Cr^0$ phases as 750 ppm (45%), 339 ppm (20%), 436 ppm (26%) and 151 ppm (9%), respectively is produced. It is notable that this distribution of secondary Cr oxide and metallic phases is largely consistent with the incomplete disproportion of $CrO_{(l)}$ upon cooling as suggested previously by Riglet-Martial et al.[15] via: $CrO_{(l)} \rightarrow 1/3Cr_{(s)} + 2/3Cr_2O_{3(s)}$. Moreover, accounting for the uncertainty arising from experimental conditions and ITFA analysis, these numbers are congruent and support previously established experimental and thermodynamic studies of Cr-doped $UO_2$[3,15] and are also consistent with the Cr−O phase diagram[22].

ITFA analysis of the Cr-doped $UO_2$ single crystal grain (Fig. 2d) revealed it to be dominated by trivalent Cr, containing 8% metallic Cr ($Cr^0$), 4% $Cr^{+2}$ and 88% $Cr^{+3}$. The $Cr^{+3}$ contribution is attributed to sole lattice incorporated Cr ($(Cr^{+3}_xU^{+4}_{1-x})O_{2-0.5x}$) as the presence of $Cr^{+3}_2O_3$ or other $Cr^{+3}$ species is ruled out by the EPR analysis discussed previously. Pertinently ITFA analysis carries a known ± 5% error[34], in which the $Cr^{+2}$ falls under, and subsequently, its presence in the crystal lattice is not supported. Moreover, contrasting the ITFA plot with the measured spectra consistently highlights the absence of characteristic peaks of $Cr^{+2}$ in the Cr-doped $UO_2$ single-crystal grains. However, the same peaks can be readily observed in the bulk material. This does contradict some previous investigations on bulk material regarding the assignment of $Cr^{+2}$ to the lattice matrix, which is not supported here. The peak at 5991 eV for the single crystal grain corresponds to the presence of the metallic Cr (found to be in the order of 8%). This is attributed to the presence of a minor amount of surface Cr metal that was not successfully removed during the crystal separation process, this effect is described and demonstrated in Supplementary Information Note 5 for this and other single crystals in a series. That the main edge peak of the Cr-doped $UO_2$ single crystal grain matches that of $Cr^{+3}Cl_3·6H_2O$ indicates the local environment and oxidation state of the Cr cation in the $UO_2$ lattice matrix and $CrCl_3·6H_2O$ are near identical. The $Cr^{+3}$ cation in $CrCl_3·6H_2O$ (space group $R\bar{3}c$) has a distorted octahedral environment with a coordination number (CN) of six to the O ligands in the hydrate. The results of the Cr K-edge HERFD-XANES measurement of the Cr-doped $UO_2$ single crystals are consistent with the previous EPR analysis, namely that $Cr^{+3}$ cation resides in a substitutional position in a distorted octahedral environment to oxygen in the $UO_2$ lattice with a neighbouring oxygen vacancy defect for charge compensation with the formula $(Cr^{+3}_xU^{+4}_{1-x})O_{2-0.5x}$. Furthermore, the HERFD-XANES measurements provide no evidence for $Cr^{+2}$ in the lattice matrix and preclude its occurrence.

To further support the redox mechanism provided by EPR analysis of $Cr^{+3}$ incorporation into $UO_2$ via oxygen vacancy defect formation as opposed to $U^{+5}$ occurrence whilst assessing any sample oxidation, HERFD-XANES measurements were performed on the U $M_4$-edge[23,35] for the Cr-doped $UO_2$ powder and U $L_3$-edge[36,37] for the Cr-doped $UO_2$ single crystal and powder using a $UO_2$ standard for both measurements. The size of the crystal coupled with absorption issues with air prevented $M_4$-edge measurements for the single crystal grain. The results are presented in Supplementary Information Note 5 Fig. 4. The measurements revealed both spectra from the Cr-doped $UO_2$ single crystal grain and powder match identically the $UO_2$, indicating the materials are purely $U^{+4}$ and there is no evidence for $U^{+5}$ in terms of edge shifting or post edge features. The sensitivity of oxidation state detection is lessened for the U $L_3$- compared to the $M_4$-edge, as the latter directly probes the $5f$ state whereas the former is the $6d$ state. Nevertheless, since the crystals are extracted from the powder which shows perfect agreement with $U^{+4}$ from the U $M_4$-edge, we accordingly do not expect a change of oxidation state between the sample types, which the $L_3$-edge measurements support, although at lower resolution comparatively. The measurements also indicate the near stoichiometric state of the investigated materials via the synthesis conditions used, particularly the absence of interstitial oxygen that would be charged balanced via $U^{+5}$. Such conditions were targeted based on the synthesis method used to inhibit hyperstoichiometry and also achieve sufficient solubility of Cr[15]. Supplementary Information Note 6 graphically illustrates these conditions via the Ellingham diagram according to Lindemer and Besman[38] for $UO_2$ with conditions used for the Cr-doped $UO_2$ preparation in this investigation. The U $L_3$ and $M_4$-edges measurements are perfectly consistent with the EPR analysis, which has enhanced sensitivity to $U^{+5}$ even at ppm concentration[28], and further reinforces the overall mechanism of incorporation via $(Cr^{+3}_xU^{+4}_{1-x})O_{2-0.5x}$ for Cr lattice incorporation following oxygen vacancy defect formation and not $U^{+5}$ formation.

## Extended X-ray absorption fine structure (EXAFS) spectroscopy

To understand the structural mechanism of $Cr^{+3}$ incorporation into the lattice structure of $UO_2$, EXAFS measurements and analysis were performed against the Cr-doped $UO_2$ powder and single crystal grains. All obtained Cr K-edge EXAFS data have been compiled in Supplementary Information Note 7 Fig. 10. The $k^3$ weighted EXAFS data show small differences between the Cr-doped powder sample and the single crystal grain. These differences are likely to arise from the compositional differences, such as the clear presence of $Cr^{+2}$ centres in the polycrystalline sample in contrast to the single crystal grains. Both samples, however, are dominated by structurally incorporated $Cr^{+3}$ in a $CrCl_3·6H_2O$-like distorted octahedral environment, which is also evident from the EXAFS data when comparing the polycrystalline Cr-doped $UO_2$ and single crystal samples with the chromium standards.

The EXAFS-data of the chromium-containing $UO_2$ samples were fitted using three different structures, $U^{+4}O_2$, $U^{+6}O_3$, $Cr^{+3}U^{+5}O_4$ as theoretical models. Cr was introduced as an absorbing and scattering atom in the uranium oxides, by replacing two uranium atoms in the cluster (see Supplementary Information Note 7 Fig. 11). The best fit for the single crystal grain was obtained using $UO_3$ as a structural model. The asymmetric $CrUO_4$ structure also yielded a rather good fit, while an untypically large $E_0$ shift was required to model the data using a Cr substituted $UO_2$ model. In all cases, the experimental spectral broadening was reduced by 1 eV in the FEFF calculations using the EXCHANGE card[39], as this sample was measured using high-energy-resolution fluorescence detection. The best description of the data was obtained when including four scattering shells in the fitting: Cr−O, a Cr−O−O multiple scattering path (MS), Cr−U and Cr−Cr. The latter scattering shell was included due to the presence of some trace non-lattice incorporated Cr species. With a slightly higher amount of $Cr^0$ in the sample (compared to $Cr^{+2}$) the Cr−Cr distance was fixed to 2.498 Å,

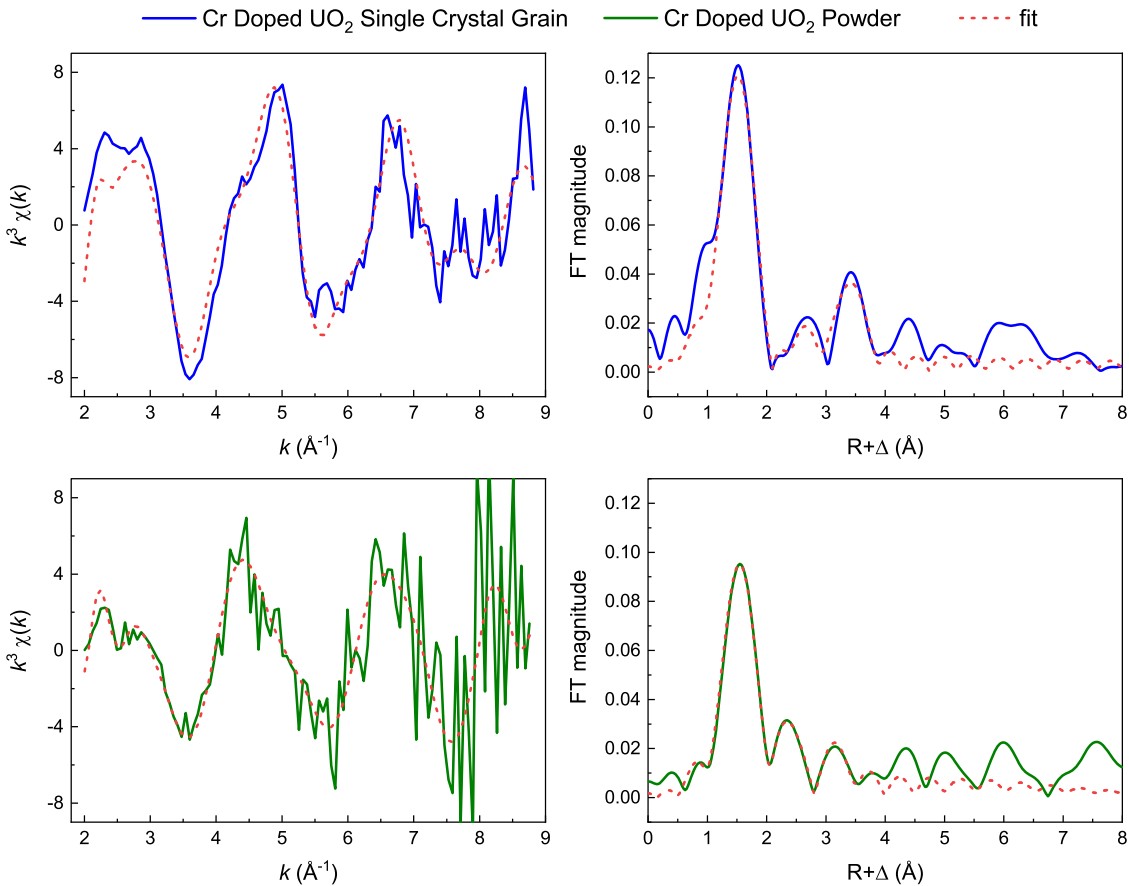

**Fig. 3 | EXAFS spectra of Cr-doped UO₂ single crystal grain and bulk material.** Fitted $k^3$-weighted Cr K-edge EXAFS spectra (left) and corresponding Fourier transforms (FT) in the spectral range from 2.0 to 8.8 Å⁻¹ (right) of the Cr-doped single crystal grain (top) and powder sample (bottom). Phase shifts (Δ) are not corrected in the FTs. The single crystal specimen was measured using HERFD mode on the X-ray emission spectrometer, while powder samples were collected in fluorescence mode using an 18-element Ge-detector.

**Table 1 | EXAFS fit parameters for Cr K-edge data measured from Cr-doped UO₂ powder and single crystal grain in addition to reference U L₃-edge EXAFS fit data from Conradson et al.[11] for UO₂**

| Sample | Shell | CN | R (Å) | σ² (Å) | Reference |
|---|---|---|---|---|---|
| Cr-doped UO₂ single crystal grain | Cr–O | 4.8 (0.8) | 1.939 (7) | 0.010[a] | Present investigation |
| | Cr–O–O (MS) | 14.4 (3.8) | 2.315 (36) | 0.010[a] | Present investigation |
| | Cr–U | 2.7 (0.7) | 3.401 (13) | 0.009 | Present investigation |
| | Cr–Cr | 0.6 (0.3) | 2.498[b] | 0.017[b] | Present investigation |
| Cr-doped UO₂ powder | Cr–O | 3.0 (0.9) | 1.984 (4) | 0.005[b] | Present investigation |
| | Cr–O–O (MS) | 12.6 (5.2) | 2.638 (7) | 0.005[b] | Present investigation |
| | Cr–U | 1.5 (1.4) | 3.487 (7) | 0.010[b] | Present investigation |
| | Cr–Cr | 2.3 (0.5) | 2.590 (7) | 0.010[b] | Present investigation |
| UO₂ powder | U–O | 7.6 (2.3) | 2.36 (2) | 0.074 (14) | Conradson et al.[11] |
| | U–U | 10.6 (2.7) | 3.88 (1) | 0.056 (7) | Conradson et al.[11] |

[a]Correlated parameters.
[b]Fixed parameters.

which is the crystallographic bond length between Cr atoms in metallic chromium[40]. The resulting fit based on the UO₃ structure is shown in Fig. 3 (top). The fit parameters are compiled in Table 1, with the EXAFS results from Conradson et al.[11] for their measurements on the U L₃-edge against UO₂ for comparison.

The Cr–O distance in the single crystal grain is approximately 1.94 Å. This is very close to the crystallographic Cr–O distance of 1.95 Å in CrCl₃·6H₂O, supporting our previous assignment of very similar local environments and oxidation states in these samples, based on the HERFD-XANES data. However, this also means that the Cr–O distance in doped UO₂ single crystals is significantly shorter (by 0.43 Å) than the U–O distance of 2.37 Å in pristine UO₂. A similar first shell contraction has been reported previously for Cr-substituted UO₂ powder[16]. This is consistent with XRD analysis in the present investigation (Supplementary Information Notes 1 and 3). The substitution of the vastly smaller Cr⁺³ cation (IR = 0.615 Å)[41] onto the U⁺⁴ site (IR = 0.95)[41] will

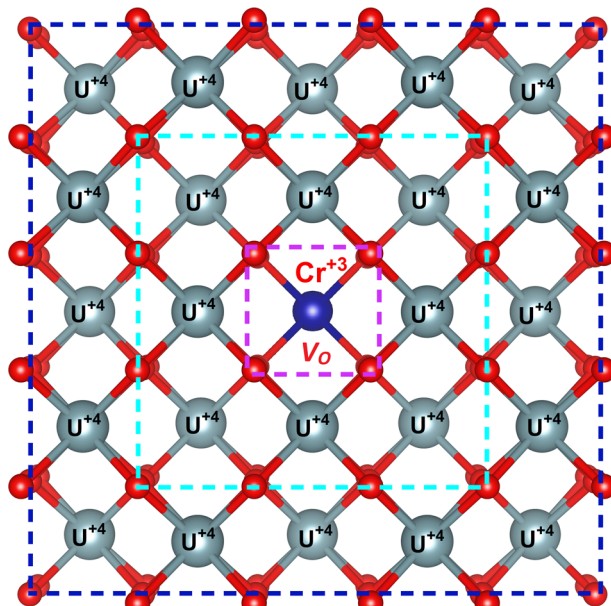

**Fig. 4 | Determined structural model for Cr lattice matrix incorporation in Cr-doped UO₂.** Structural representation of the box-shell incorporation model of the incorporation of Cr into the UO₂ lattice matrix forming $(Cr^{+3}_xU^{+4}_{1-x})O_{2-0.5x}$ as derived from the EPR, HERFD-XANES and EXAFS analysis of this investigation. The 1st box-shell (purple) highlights the $Cr^{+3}$ cation incorporation which involves the formation of an oxygen vacancy defect, $V_O$, for charge compensation. The 2nd box-shell (cyan) indicates the locally distorted U lattice region derived from EXAFS analysis which showed shorter Cr–U distances resembling an environment closer to UO₃ or CrUO₄ than that of U–U in pristine UO₂. The 3rd box-shell (dark blue) which resembles closely that of the $U^{+4}O_2$ lattice, i.e., far enough away from the $Cr^{+3}$ cation to not receive significant lattice distortions. Furthermore, the uranium cationic lattice stays consistently as $U^{+4}$.

likely result in a contraction of the metal-to-oxygen bond distance, which is reflected in the EXAFS analysis. This reduction in metal-to-oxygen bond distance would also induce local distortion, which is consistent with the previous EPR analysis. Interestingly, a very similar difference of 0.47 Å between the obtained Cr–U bond length and the crystallographic U–U distance can be derived, implying that the considerable distortions to the crystal lattice upon the incorporation of the small trivalent chromium cation are not dampened for the second coordination shell. With these rather large differences between the local environments of the dopant and the host cations, it is clear that the theoretical calculations using UO₂ as input structure did not yield a satisfactory description of the measured EXAFS data. In comparison, both UO₃ and CrUO₄ have Cr/U–O distances like those we obtained for our single crystal grain, namely 2.08 and 2.02–2.05 Å, respectively.

Ideally, the charge compensation mechanism derived by the EPR method, involving oxygen vacancy formation following Cr incorporation into the UO₂ lattice can be corroborated by the obtained coordination numbers (CNs) from our EXAFS fits. As indicated in Table 1, the CNs are rather low in comparison to the expected values of 8 and 12 for the first and second coordination shells in UO₂, respectively. The samples were not diluted prior to measurement, implying that self-absorption effects may reduce the signal amplitudes and thereby the CNs. In addition, lower CNs are typically obtained for systems with a very heterogeneous distribution of interatomic distances and accompanying destructive interference effects[42]. Thus, although the CNs are lower with respect to the crystallographic host structure, definite conclusions on vacancy formation for charge compensation cannot be drawn from our EXAFS data. Nevertheless, the EPR analysis with the U $M_4$- and $L_3$-edge measurements, as discussed previously, provides a definitive description of the incorporation mechanism, particularly

considering the superior sensitivity of EPR for trace amounts and the chemistry of $Cr^{+3}$ and $U^{+5}$. A structural representation based on the EPR, HERFD-XANES and EXAFS analysis for the incorporation of $Cr^{+3}$ into UO₂ is provided in Fig. 4 which illustrates the structural different "box" shells present around the $Cr^{+3}$ cation in the UO₂ lattice matrix.

Finally, attempts were made to fit the EXAFS data collected for the Cr-doped UO₂ powder sample. The signal-to-noise ratio of the collected data is rather poor, and the chromium chemistry in the sample is very complex, as described previously by the ITFA analysis from the HERFD-XANES. Using the same fitting approach as previously applied for the single crystal grain only yielded a satisfactory fit when allowing individual $E_0$ shifts for the different scattering shells. This is justified when the sample is composed of several different phases, which cannot be described by one structural model only. With almost a third of the Cr in the UO₂ powder sample consisting of metallic chromium and $Cr^{+2}$ the EXAFS fit for variable $E_0$ shifts was acquired. The Cr–O and Cr–U distances are very similar to the single crystal grain. Assuming that the $Cr^{+2}$ in the UO₂ powder sample (26%) has a CrO-like environment with a crystallographic Cr–O bond length of 2.08 Å, no large changes in the first shell distance in comparison to the Cr–O distance in substituted UO₂ (1.94 Å, derived from the single crystal grains) are expected. The Cr–Cr bond length in the powder sample is clearly longer than in the single crystal grain. This can be understood when considering the crystallographic Cr–Cr distance in CrO (2.942 Å) in comparison to Cr metal (2.498 Å). The strong similarity between the $Cr^{+3}$ nearest and next nearest neighbour distances in both Cr-doped UO₂ powder and single crystal grains to UO₂ strongly supports the assignment that the incorporation of Cr into the lattice involves a substitutional mechanism for U. We note further the strong consistency between the distribution of different Cr bearing phases present based on the EXAFS analysis, which matches that of the HERFD-XANES (described previously), where both support previously thermodynamic measurements and estimates of Cr-doped UO₂[15].

## Discussion

The combined use of EPR, HERFD-XANES and EXAFS on Cr-doped UO₂ powder and extracted single crystal grains has demonstrated that Cr is incorporated into the UO₂ lattice matrix substitutionally for a $U^{+4}$ position as $(Cr^{+3}_xU^{+4}_{1-x})O_{2-0.5x}$ with the additional formation of $Cr^0$, $Cr^{+2}$ and $Cr^{+3}$ states that are found within bulk material, but not in the lattice matrix. It is important to consider the origin of these non-lattice incorporated $Cr^0$, $Cr^{+2}$ and $Cr^{+3}$ species identified in the HERFD-XANES measurements of the bulk material but absent in the single crystal grains. It has been previously established that under high-temperature sintering conditions (1500 °C < T < 2000 °C, −460 kJ/mol < $\mu_{O2}$ < −360 kJ/mol), used in the present study, other similar investigations[3,6,10,15,22,24] and during industry fabrication[2,5], Cr can readily adopt $Cr^0_{(s)}$, $Cr^{+2/+3}_3O_{4(s)}$ $Cr^{+2}O_{(l)}$ and $Cr^{+3}_2O_{3(s)}$ states. Their quantities will depend on specific synthesis conditions used and will also be affected via the cooling pathway according to the phase diagram of Cr–O[22]. At high temperatures, liquid $Cr^{+2}O$ is suspected of strongly contributing to the enhanced grain growth along with $Cr_2O_3$ in Cr-doped UO₂ and evidence of this process has been identified in previous investigations[15,24]. However, $Cr^{+2}O$ is not stable at ambient conditions[22] and considering the slow cooling rates used for Cr-doped UO₂ synthesis in the present study (6 °C/min), the $Cr^{+2}$ identified by HERFD-XANES is not attributed to quenched $Cr^{+2}O$. Leenaers et al.[3] identified using electron probe microanalysis non-lattice associated Cr–O precipitates with compositions close to CrO stoichiometry, attributing them to reduced $Cr_2O_3$. Indeed, substoichiometric $Cr_2O_3$ has been previously rationalised through the occurrence of $Cr^{+2}$ in previous studies of chromia[43–45]. Riglet-Martial et al.[15] observed similar Cr:O ratios in Cr precipitates/enriched regions to that of Leenaers et al.[3], approaching stoichiometries of CrO which they attributed to partial and incomplete disproportionation of liquid $Cr^{+2}O$ by:

$CrO_{(l)} \rightarrow 1/3Cr_{(s)} + 2/3Cr_2O_{3(s)}$. When the quantitative results of the ITFA HERFD-XANES analysis of the bulk materials secondary Cr phases are considered as discussed earlier, these appear consistent with proposed incomplete disproportionation. $Cr^{+3}{}_2O_{3(s)}$ and $Cr^0$ are readily observed from HERFD-XANES/EXAFS and SEM-EDS measurements in the present investigation, in addition to several previous studies[3,10,15,16,21]. Recent high resolution transmission electron microscopy analysis of Cr-doped $UO_2$ has further identified the significant enrichment of Cr and structural disorder found in precipitate grain boundary regions[46]. Further recent studies by Devillaire et al.[47] found evidence for $Cr_3O_4$ in these precipitates via Raman spectroscopy, which would involve $Cr^{+2}$. It has also been proposed the possibility of $Cr^{+2}$ occurring within the $UO_2$ lattice matrix itself with some redox mechanisms between it and $Cr^{+3}$ which should be observable at ambient conditions.[8,48] We note relevant studies to this argument have only investigated bulk material at ambient conditions which carries significant deconvolution in Cr measurement, as demonstrated by Fig. 2b and c regarding the difference between bulk and lattice structure, respectively. Analysis of extracted single crystal grains that were prepared under reducing conditions (1700 °C and $\mu_{O2}$ of −420 kJ/mol) and were subject to even more reducing conditions during slow cooling, since the atmosphere was maintained resulting in a reduction of $\mu_{O2}$ with cooling, provide no evidence for the occurrence of +2 or possible redox mechanisms enabling it on measured material, and subsequently it is precluded from the lattice matrix. We further note with respect to literature, this study provides the most sensitive and highest resolution spectroscopic measurement via EPR and Cr K-edge HERFD-XANES respectively to Cr-doped $UO_2$, in which neither technique has been applied to Cr-doped $UO_2$ prior to this investigation, nor have single crystal grains been previously examined. Moreover, considering observations and analysis from the present investigation and literature, it appears the origin of $Cr^{+2}$ observed in ambient condition bulk measurements, is related to sub-stoichiometric enriched Cr precipitate-grain boundary regions as proposed previously by Riglet-Martial et al.[15] and Leenaers et al.[3]. Other Cr states might interact with the $UO_2$ lattice matrix at elevated temperatures during material preparation or reactor operation conditions according to Cooper et al.[19]. However, when the results of Mieszczynski et al.[16], who examined Cr-doped $UO_2$ under fresh and reactor irradiated conditions using a different preparation route, are compared against those of the present investigation, they show consistency when the non-lattice Cr phases are accounted for. Consequently, by comparing the results of the present investigation against others, such as Mieszczynski et al.[16] the chemical state of lattice substituted Cr in $UO_2$ appears ubiquitously as +3, via $(Cr^{+3}{}_xU^{+4}{}_{1-x})O_{2-0.5x}$, even when considering pre- and post-reactor irradiated material. Additionally, the secondary phase Cr species: $Cr^0$, $Cr^{+2}$ and $Cr^{+3}$ which are not observed in single crystal lattice grains, are attributed to Cr ($Cr^0$), $Cr^{+2}$ and $Cr^{+3}{}_2O_3$ found in enriched Cr grain boundary and precipitate regions of bulk material. The proportions of these are dependent upon sintering conditions used as described well in previous thermodynamic and experimental studies[3,15].

EPR, HERFD-XANES and EXAFS spectroscopic analysis of Cr-doped $UO_2$ bulk powder and extracted single crystal grains has demonstrated the Cr cation incorporates into the $UO_2$ lattice matrix by substitution of the $U^{+4}$ cation as $Cr^{+3}$, with the additional formation of oxygen vacancy defects, $(Cr^{+3}{}_xU^{+4}{}_{1-x})O_{2-0.5x}$. The occurrence of $U^{+5}$ is discounted by EPR in addition to U $M_3$-edge and U $L_3$-edge measurements of both Cr-doped $UO_2$ bulk powder and extracted single crystal grains, providing no evidence for it. Furthermore, the EPR spectra can only be explained via the occurrence of a lone $Cr^{+3}$ cation with oxygen vacancy defects, where other incorporation mechanisms such as via neighbouring twin $Cr^{+3}$ cations with a neutral vacancy, or a lone $Cr^{+2}$ cation are ruled out. The additional Cr states quantitatively identified via HERFD-XANES with subsequent ITFA analysis, that were absent in

the single crystal grains but present in the bulk material, include metallic Cr ($Cr^0$), $Cr^{+2}$ and $Cr^{+3}{}_2O_3$, are attributed to be occurring in precipitate and grain boundary regions. It is argued that some previous experimental investigations of bulk Cr-doped $UO_2$ have often erroneously concluded the Cr lattice chemical state based on the occurrence of these secondary Cr oxide and metallic species that interfere with bulk measurement leading to equivocal results. The extraction of representative single crystal grains ameliorates the issue of additional Cr species interfering in measurements and allows the Cr chemical state in the lattice matrix to be measured directly, free of contaminant signal. By comparing the results against the literature, the proportions of secondary oxide and metallic chromium are argued to be dependent on synthesis conditions[6], particularly oxygen potential and temperature, whereas the lattice matrix $Cr^{+3}$ state appears ubiquitous even after reactor irradiation[16]. The results of the investigation are further consistent and appear to corroborate previous thermodynamic measurements and estimates of Cr-doped $UO_2$[3,15]. Consequently, this experimental investigation has incontrovertibly established the mechanism of lattice incorporation for Cr and its associated secondary chemical states that occur in Cr-doped $UO_2$ nuclear fuels.

## Methods

### Synthesis

Cr-doped $UO_2$ material was generated using a co-precipitation method beginning with doped ammonium diuranate (ADU) based on a previously established method[6] for the production of samples that are representative of those generated by industry. The doped ADU was prepared using stoichiometrically controlled amounts of uranyl nitrate ($UO_2(NO_3)_2$) and chromium nitrate ($Cr(NO_3)_3$) that were mixed in solution, followed by precipitation using ammonia ($NH_3$) involving a 300% excess. Samples with targeted 0 and 3500 ppm additions of $Cr_2O_3$ (2395 ppm as Cr elemental) were generated via this method. Supersaturated Cr conditions, above the predicted solubility limit of 750 ppm of Cr at 1700 °C and $\mu_{O2}$ of -420 kJ/mol[15], were chosen to ensure maximum incorporation of Cr into the matrix in spite of the loss of Cr from potential volatilisation[24], and subsequently enable optimum measurement signal resolution, considering the small size of single crystal grains. The ADU mixtures were firstly calcined under air to oxide form using a box furnace at 800 °C for 5 h prior to reduction to $UO_2$ at 600 °C under 4% $H_2$−96% Ar atmosphere for an additional 5 h. In both steps, the samples were in powder form. Finally, the pre-treated powders were compacted into pellets and heated to 1700 °C using a tube furnace for 10 h with a $\mu_{O2}$ of −420 kJ/mol that was monitored and produced via a 4% $H_2$−96% Ar with 1% $H_2$−99% Ar gas mixture. The furnace was cooled to room temperature using a ramp rate of 6 °C/min whilst the gas mixture was maintained. Post-synthesis, samples were carefully mechanically divided for specific measurement analysis.

### Scanning electron microscopy

The morphology, grain size and average distribution of Cr were determined using an FEI Quanta 200F environmental scanning electron microscope (ESEM) fitted with an energy-dispersive X-ray spectrometer (EDS). Surfaces of parts of the sintered pellets were embedded in resin and carefully polished using diamond pastes (up to 1 μm) and finished with a colloidal Si-suspension. The samples were sputtered with a thin layer of carbon to enhance the electrical conductivity. Backscattered electron (BSE) images were collected at 10 kV to obtain a high-orientation contrast image to identify single grains. The procedure of Podor et al.[49] was used for image acquisition, a semi-automatic segmentation of grain boundaries and the determination of grain sizes. EDS spot analyses as well as EDS elemental mappings, to determine the spatial distribution of Cr, were carried out at 20 kV and a working distance of 10 mm. Single crystal grains were measured neat pre-EPR, HERFD-XAS and EXAFS measurements to check crystal purity

and particularly the absence of surface and subsurface Cr oxide impurities and contaminants.

## Synchrotron X-ray powder diffraction

Ambient temperature synchrotron X-ray powder diffraction (SXRD) experiments were performed at the BM20 Rossendorf beamline[50] (ROBL) at the European Synchrotron Radiation Facility (ESRF), Grenoble, France. Diffraction data were collected on a high-resolution XRD1 machine equipped with a Dectris Pilatus 100k photon counting detector. Synthesised samples were finely ground and packed in glass capillaries of 0.3-mm diameter enclosed in 1-mm Kapton tubes that serve as confinement barriers. The energy of synchrotron radiation was set at 16,000 eV and the geometry of the experimental setup was determined using a NIST LaB$_6$ standard reference. Experiments were performed in transmission mode and corresponding 2D data were reduced using the PyFAI library adapted for diffractometers mounted on a goniometer arm[51]. Structural analysis was performed using the Rietveld method as implemented in the programme GSAS-II[52]. The peak shapes were modelled using a pseudo-Voigt function and the background was estimated using a 6–12 term shifted Chebyschev function. The scale factor, detector zero-point and lattice parameters were refined together with the peak profile parameters. All measured samples were determined to be single-phase fluorite in space group $Fm\bar{3}m$ from Rietveld refinement analysis (Supplementary Information Note 1).

## Single crystal X-ray diffraction and mechanical separation of single crystal grains

It was determined from SEM analysis that single crystal grains would be of appropriate size for mechanical separation in the Cr-doped UO$_2$ sample (Supplementary Information Note 2 Table 1). Accordingly, single crystal grains were mechanically separated from the synthesised material by breaking the pellets, firstly by fracturing them using a mortar and pestle then cutting them under a microscope to separate individual crystals whilst sonicating them to assist in removing surface impurities. Note, single crystal grains were separated immediately after synthesis i.e., not subjected to SEM polishing or other chemical treatments. Single crystal X-ray diffraction (SC-XRD) measurements were performed on single crystal grains using an Agilent Oxford Diffraction Super Nova diffractometer with a Mo Kα tube at 296 K, equipped with CrysAlisPro software. The unit cell was determined, and background effects were processed by the CrysAlisPro software. The initial structures were refined using SHELXL-2018 within the WinGX (v1.80.05) software[53], and the ADDSYM algorithm of the PLATON programme[54] was used for the checking of possible higher symmetries. Absorption corrections for the raw data were performed using the multiscan method. Samples of appropriate single crystal quality from SC-XRD analysis i.e. confirmation of the UO$_2$ fluorite structure and to the limits of resolution free of twinning, contaminant material or incorrect reflections were separated for later analysis. No attempts were made to determine the position of Cr atoms within UO$_2$ crystals from SC-XRD measurements. Results for SC-XRD measurements and analysis are provided in Supplementary Information Note 3.

## Electron paramagnetic resonance spectroscopy

Electron paramagnetic resonance spectroscopy was carried out on an X-Band CW-EPR spectrometer system of the ELEXSYS E500 Series, with a 10" magnet and 12 kW power supply (Bruker Biospin, Rheinstetten, Germany) equipped with an Oxford Instruments Mercury iTC He cryostat. The measurements were performed against five Cr-doped UO$_2$ crystals recorded at ambient temperature with 9.85 GHz and 20.00 mW microwave power, 1.5 mT modulation amplitude and 100 kHz modulation frequency. An empty EPR tube was recorded using the same conditions. The obtained signal was used as background and subtracted from the Cr-doped UO$_2$ sample. The data could

be fitted using EasySpin[55] including one species with $S = 3/2$. Cr$_2$O$_3$ was also measured for reference with EPR and the results provided in Supplementary Information Note 4.

## High energy resolution fluorescence detection X-ray absorption near-edge spectroscopy

Cr K-edge, U M$_4$-edge and L$_3$-edge HERFD-XANES measurements were performed at the BM20 ROBL[50] beamline of the ESRF, in Grenoble, France. The incident energy was scanned using a Si(111) monochromator. HERFD-XANES spectra were collected at room temperature using an X-ray emission spectrometer equipped with five Ge(422) crystal analysers with 1 m bending radius, and a silicon drift X-ray detector in a vertical Rowland geometry[56]. For Cr K-edge HERFD-XANES measurement, the spectrometer was tuned to the maximum 5.4149 keV X-ray emission line using the 422 reflections of a Ge analyser at a Bragg angle of 82°. For U M$_4$-edge HERFD-XANES measurements, the spectrometer was tuned to the maximum U Mβ ($3d3/2-4f5/2$, 3337 eV) X-ray emission line using the Si (220) reflection with a Bragg angle of 75°. For U L$_3$-edge HERFD-XANES measurements, the spectrometer was tuned to the maximum U Lα$_1$ ($2p3/2-3d5/2$, 13,614 eV) X-ray emission line using the Si (880) reflection with Bragg angle of 71.5°. The detected intensity was normalised to the incident flux for both L$_3$ edges. Beam size was estimated to be 30 μm (vertically) by 1000 μm (horizontally). An energy resolution of 1.1 eV was obtained via measurement of the full-width half maximum of the elastic peak at the incident energy of 5.4149 keV. Cr-doped UO$_2$ powder and single crystal grains with 3500 ppm addition as Cr$_2$O$_3$ were measured on the Cr K-edge in addition to the Cr standards: Cr$^{+2}$Cl$_2$, Cr$^{+3}$Cl$_3$·6H$_2$O, Cr metallic and Cr$^{+3}_2$O$_3$. For Cr-doped UO$_2$ powder and single crystal grains, the former was mounted without a backing matrix and the latter as neat monoliths. Cr-doped UO$_2$ powder was measured with UO$_2$ on the U M$_4$-edge and U L$_3$-edge. Due to the sensitivity towards oxidation of CrCl$_2$ it was prepared for measurement using an Ar-filled glovebox containing less than 1 ppm of O$_2$/H$_2$O where the reference was confined using a double-layer system to prevent air intrusion during measurement. Data analysis was performed by using the software package ATHENA[57] and iterative transformation factor analysis (ITFA)[34].

## Extended X-ray absorption fine structure spectroscopy

EXAFS measurements were conducted on a selected Cr-doped single crystal sample using high energy resolution fluorescence detection (HERFD) mode and a X-ray emission spectrometer as explained above. The validity of HERFD-EXAFS was confirmed by analysis of conventional and HERFD EXAFS of the Cr$^{+3}$Cl$_3$·6H$_2$O standard. Additional measurements of chromium oxidation state standards (Cr foil, CrCl$_2$, CrCl$_3$·6H$_2$O), as well as the Cr-doped UO$_2$ bulk powder, were performed in fluorescence mode using an 18-element Ge-detector. A minimum of three spectra were collected for each sample, followed by dead-time correction and energy calibration using EXAFSPAK[58]. Data reduction was performed with the programme SIXPACK, while Fourier transforms (FT) of the EXAFS data over a constant $k$-range (2.0–8.8 Å) and subsequent fitting was conducted with the programme WinXAS version 3.2[59]. Theoretical scattering phases and amplitudes were calculated using the ab initio code FEFF8.20[60]. Several theoretical models were used as input for the fitting of the EXAFS data. The models were based on structures for UO$_2$ and UO$_3$ with Cr substitution for the host cation site. In addition, chromium monouranate (CrUO$_4$) was used as a model for a Cr$^{+3}$–U$^{+5}$ environment in the substituted UO$_2$ matrix. The clusters are compiled in Supplementary Information Note 7, Fig. 11.

## Data availability

The experimental EPR data (Fig. 2a), HERFD-XANES data (Fig. 2b), and EXAFS data (Fig. 3) that support the findings of this study have been deposited in the data repository RODARE under the link https://doi.org/10.14278/rodare.2196.

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

## Acknowledgements

The authors, G.L.M., S.G., V.S., J.M., T.L., C.H. and N.H., are grateful to funding and support from the German Federal Ministry of Education and Research (BMBF), Project No. 02NUK060, that enabled this research. Dr. Juliane März is kindly acknowledged for her help collecting single crystal grains for the EPR measurements. The ESRF Chemistry Laboratory personnel Harald Muller and Léa Bourcet are acknowledged for their help with the glove-box facility. Dr. Andrey Bukaemskiy is kindly acknowledged for assistance in pellet pressing and polishing. Dr. Gregory Leinders is kindly thanked for fruitful discussions on the thermodynamics of $UO_2$.

## Author contributions

The project was conceived and developed by G.L.M. The research methodology, experimental planning, and formal analysis was jointly conducted by G.L.M. and N.H. The materials were synthesised by G.L.M., P.K. and M.H. Single crystal grains were extracted and characterised with single crystal X-ray diffraction by G.L.M. Synchrotron powder diffraction measurements and analysis were performed by G.L.M., V.S. and C.H. Electron paramagnetic measurements and analysis were performed by R.G., S.G., P.K. and N.H. SEM measurements and analysis were made by R.T. and M.K. X-ray absorption spectroscopy measurements were performed by G.L.M., N.H., J.M., T.L., A.R., E.F.B. and K.O.K. Manuscript writing, review and editing was performed by G.L.M. and N.H. with input from all authors.

## Funding

## Competing interests

The authors declare no competing interests.
