## [Peer Review File · Nature Communications]

Deconvoluting Cr States in Cr-Doped UO₂ Nuclear Fuels via Bulk and Single Crystal Spectroscopic StudiesREVIEWER COMMENTS

Reviewer #1 (Remarks to the Author):

The authors present a very interesting piece of work investigating the charge states and accommodation mechanisms for Cr in doped UO₂. Both single crystal data and powder are assessed to elucidate the difference between Cr accommodated in the UO₂ lattice vs that maintained as separate phases at the grain boundaries. The key results presented by the authors are that (for the conditions examined) Cr in the UO₂ lattice substitutes for U atoms as Cr(III), whereas Cr outside the bulk lattice forms precipitates of Cr or CrO. This work is of strong interest to the nuclear fuels community, given that the exact location and nature of Cr in doped UO₂ is critical to develop understanding of how Cr affects fuel performance and fabrication. It is also of interest to the wider materials community. Several outstanding questions should be addressed:

1) In figure 1 it is not clear to me that precipitates are only observed at the grain boundaries. In the BSE image of polycrystalline sample, it seems as though there are plenty of precipitates within the grain interior. Similarly the Cr and O map seem to indicate precipitates not only lying on the grain boundaries. By overlaying the grain boundary structure on top of the Cr map it would be much clearer whether or not this is the case. If precipitates are seen in the grains in figure 1, it would contradict several statements in the text that all Cr³⁺ identified in the single crystal, is due to Cr in the UO₂ lattice rather than in Cr₂O₃ intra-granular precipitates. Similarly, the single crystal images in figure 1 show that no precipitates remain on the surfaces but do not prove that precipitates do not exist within the single crystals.

2) Related to point 1, it is plausible that under sintering conditions Cr went into solution in the UO₂ lattice but then a significant proportion precipitated out upon cooling. The author should make clear that the results they are showing refer to the end state of doped UO₂ at RT, following potential redistribution of the dopant either between different possible accommodation sites or through precipitation, as well as possible changes in charge state. This should also be discussed in the context of nuclear fuel, which operates at temperatures much greater than that which these experiments were carried out.

3) In the introduction, the authors mention that dopants improve the power output of the fuel. This is counter intuitive to me as I would expect any dopant to lower the U-loading, and also potentially act as a neutron absorber, therefore reducing the power output. In the same sentence the authors suggest thermal properties are improved - are they referring to thermal conductivity or other properties like specific heat?

4) The statement "This can be interpreted as a high-spin Cr⁺³ together with oxygen vacancy defects in the vicinity of the substituted Cr site to balance the charge," on page 5 line 104. Do the author mean that this is the only way the data can be interpreted or is this one of several ways it could be? It would be beneficial if the authors list in a bit more detail other defect complexes and charge states that they considered but ruled out.

5) It is mentioned that Ln³⁺ doping is charge compensated by U⁵⁺, unlike Cr³⁺, which the authors suggest is compensated by oxygen vacancies. There must be a solid physical explanation for this difference? To my mind, Ln³⁺ and Cr³⁺ could, in theory, be charge compensated either way depending on the oxygen potential. The preference would mostly be governed by $V_{O} + 0.5O_2 + 2U^{4+} \rightarrow 2U^{5+}$, whereby, if the oxygen potential is lowered, V_{O} will become dominant. Differences between the dopants could arise due to binding between the dopant and either V_{O} or U⁵⁺. Do the authors think that the Cr-doped fuel has a lower oxygen potential than for the Ln-UO₂ experiments or do they believe that the dopant- V_{O} and dopant-U⁵⁺ binding energies are dramatically different?

6) I would like the authors to elaborate more on why they believe the proportion of Cr in the 2+ state is so certainly in a second phase, given that CrO does not appear on Cr-O phase diagram at these temperatures. As far as I am aware, CrO forms as a liquid at sintering temperatures but should be converted to Cr metal and Cr₂O₃ when cooled. I understand the argument that the lack of Cr²⁺ in the single crystal data indicates that it is not in the bulk lattice but, if the CrO

precipitate argument is invoked, it should be explained how this does not contradict the phase diagram. Could Cr²⁺ be incorporate as individual atoms within the grain boundary and then get chipped off during mechanical processing?

7) To what accuracy are the authors certain their undoped UO₂ sample is exactly stoichiometric? Figure 3 is used to show that additions of Cr do not increase the U⁵⁺ concentrations, thereby indicating Cr is not charge compensated by U⁵⁺. However, if there is even a very small amount of non-stoichiometry in UO_{2+x}, approximately x=0.001, the U⁵⁺ concentration would be governed by the concentration of the oxygen interstitials, rather than by the potentially even smaller quantity of Cr in solution. In this case, Cr could still be charge compensated by U⁵⁺ but it would have a negligible impact on the overall U⁵⁺ concentration. I did these estimates pretty roughly but the authors should demonstrate that this possibility does not affect their conclusions.

One very minor comment: figure 2 seems to be a jpeg - it would look better as png or pdf.

Reviewer #2 (Remarks to the Author):

The work by Murphy et al. seeks to resolve the discrepancy in the literature regarding the oxidation state of Cr in UO₂. The topic is of importance for the nuclear community and the use of EPR on this class of materials is novel to the best of the reviewer's knowledge. Though, the more important information is how the Cr facilitates improved properties during sintering, which would require in situ studies rather than measurements under ambient conditions. The work may be better suited to a nuclear materials focused journal. That said, the methodology and conclusions drawn are overall logical and appropriate. Below are a few comments and questions which should be addressed/considered.

Page 2: line 46, 47, in fact, the solubility of Cr into UO₂ is subject to debate. Some claim a solubility limit as high as double what the authors claim here [Cardinaels et al. Journal of Nuclear Materials 424. (2012) 252-260]

Page 3 line 74 No argument that the extracted single crystals were completely representative of the material overall, but what's the basis for this claim. How many extractions performed and how do they compare to one another and the bulk material structural and electron spectroscopic measurements? It appears only one single crystal diffraction pattern was analyzed, and the unit cell parameters are quite different than that of the bulk.

Page 6 line 127 what is the energy resolution of the HERFD-XANES measurements performed here?

Page 9 line 177 "... compared to the single crystal grain The spectrum..." missing period between sentences.

Page 9 line 190 "Figure (c) and (d)" needs figure number as well. I presume figure 2c and 2d.

Which single crystal was diffraction performed on (SC1 or SC2)? If both were measured, how do they compare to one another and to the bulk powder?

Valid that the EPR data shows no evidence of U⁵⁺, but not convinced that HERFD-XANES possesses the sensitivity to corroborate these results. The signal is dominated by U⁴⁺ and lacks resolution to observe a small shift associated with ppm levels of U⁵⁺.

Reviewer #3 (Remarks to the Author):

The paper addresses the oxydo-reducing properties of chromium within doped UO₂ nuclear fuel. It

aims at identifying the valence state(s) of chromium within the material, in particular in the fluorite-type UO₂ phase, using advanced spectroscopic techniques.

Assessment of chromium oxide as dopant for UO₂ system is nowadays more than ever of great current interest for regulator and fuel vendor. Much research is in progress and still needed to clarify the role of additives in UO₂ fuel as regards the material properties (gas transport, precipitation or release of fission gas, mechanical behaviour, thermal conductivity, etc.). The aim is to design nuclear fuel material showing safe behaviour (Accident Tolerant Fuels) as well as optimised performances (limitation of fission gas release, prevention of clad failure by I-CSC, etc.) in both normal and off-normal operating conditions.

As such, the topic addressed in this article is of prime current interest and worthy for publication in Nature Communications.

It is well known, from a number of papers available in the open literature, that chromium oxide is very sensitive to oxido-reduction in the conditions relevant for fuel fabrication and nuclear LWR fuel operation.

As such :

- the speciation and solubility of chromium in UO₂ is very much dependant on both temperature and oxygen partial pressure, which may result in a great variety of UO₂ fuel microstructures;
- the behaviour of Cr-doped fuel in operation (normal and off-normal conditions) is essentially determined by the oxido-reducing conditions (pO₂ versus T) taking place within the fuel pellets.

The complexity of the chemical speciation of chromium in conditions relevant for nuclear fuels as well as its high sensitivity to redox variations require great experimental caution in the dedicated experimental work.

In the present study, the scientific work is well conducted, with great experimental care, and give rise to high quality data. In contrast to previous investigations, special attention is paid to physically separate the Cr-doped UO₂ matrix crystals from the chromium precipitates in separate phase, in order to overcome the difficult problem of convolution of the spectroscopic signals and to identify unambiguously the chemical states of chromium within the material.

Some minor points to consider in the final version:

1. As solubility of chromium in UO₂ varies as a function of temperature and oxygen potential, it is essential to systematically explain the (T, pO₂) conditions related to the specified solubility values. In addition, it is also essential to specify whether the given solubility values refer to ppm Cr in UO₂ or ppm Cr₂O₃ in UO₂ (which is quite different).
2. In the preparation section, the amount of chromium added to the samples need to be more clearly defined : Is it "addition of 3500 ppm Cr" or "addition of Cr corresponding to 3500 ppm Cr₂O₃ in UO₂" in consistency with Kugler et al. previous study ?
3. The word "purity" is used in order to state that the single crystal is devoid of secondary chromium bearing phases, but it may lead to confusion with regards to the chemical purity of UO₂ itself (without any ppm of Ca or Fe...). Clarification needed.
4. The recent publication <https://doi.org/10.1038/s42004-022-00784-3> (December, 1st, 2022) could be additionally discussed in order to explain whether the observed differences in the oxidation state of chromium come from irrelevant thermodynamic conditions (uncontrolled O₂ partial pressure, more reducing conditions?) or from secondary phases bearing chromium (quenched liquid CrO ?) which were too much represented in the sample.
5. In figure 5, the only possibility given is a Cr³⁺ with a neighbouring oxygen vacancy with charge +1. Couldn't it be two neighbouring Cr³⁺ with a neutral oxygen vacancy? Would this latter case lead to a hyperfine structure in Cr³⁺ EPR spectrum?
6. The consistency between the present spectroscopic observations and thermodynamics should be discussed more extensively.
7. Two references requires completion.
8. Minor typographic inaccuracies require correction.

Extended comments are included in the enclosed files.

Reviewer #4 (Remarks to the Author):
Prepared report with Reviewer #3.

Reviewer Response

Reviewer #1 (Remarks to the Author):

The authors present a very interesting piece of work investigating the charge states and accommodation mechanisms for Cr in doped UO₂. Both single crystal data and powder are assessed to elucidate the difference between Cr accommodated in the UO₂ lattice vs that maintained as separate phases at the grain boundaries. The key results presented by the authors are that (for the conditions examined) Cr in the UO₂ lattice substitutes for U atoms as Cr(III), where as Cr outside the bulk lattice forms precipitates of Cr or CrO. This work is of strong interest to the nuclear fuels community, given that the exact location and nature of Cr in doped UO₂ is critical to develop understanding of how Cr affects fuel performance and fabrication. It is also of interest to the wider materials community. Several outstanding questions should be addressed:

1) In figure 1 it is not clear to me that precipitates are only observed at the grain boundaries. In the BSE image of polycrystalline sample, it seems as though there are plenty of precipitates within the grain interior. Similarly the Cr and O map seem to indicate precipitates not only lying on the grain boundaries. By overlaying the grain boundary structure on top of the Cr map it would be much clearer whether or not this is the case. If precipitates are seen in the grains in figure 1, it would contradict several statements in the text that all Cr³⁺ identified in the single crystal, is due to Cr in the UO₂ lattice rather than in Cr₂O₃ intra-granular precipitates. Similarly, the single crystal images in figure 1 show that no precipitates remain on the surfaces but do not prove that precipitates do not exist within the single crystals.

Response: We thank the Reviewer for highlighting the limitations in the previous descriptive text and Figure. Indeed, the Reviewer is correct that precipitates are occurring on the surface of the grains, consistent with previous investigations such as by Mieszczyński et al. and Bourgeois et al. The Figure has, thereby been updated as suggested by the Reviewer, and the text has been revised accordingly. However, we would like to highlight, that the bulk material and the single crystal grains are different in terms of detectable impurities. As pointed out by the Reviewer, we agree the EDS is limited to the surface and near subsurface on element detection. However, comparing the updated Figure 1 in the revised manuscript with the bulk material and single crystal grain, it shows that detectable surface species that can be readily observed on the bulk material are clearly absent in the single crystal grain despite the x20 relative magnification of the single crystal grain compared to the bulk material. We re-analysed our SEM-EDS measurements of the bulk material and identified that Cr-O spots on the grains are associated with void regions in the grains, these voids can be readily observed in the single crystal grain in Figure 1. Any surface impurities present, particularly in these voids, on the single crystal grains should thereby be clearly visible in the EDS maps, but inspecting Figure 1 highlights their absence. Nevertheless, the Reviewer is correct that in the previous version it does not rule out sub-surface impurities sufficiently in the text used. A key part of our investigation is the use of EPR, which we applied to the crystal in Figure 1 among many others. EPR has a detection limit far superior to that of EDS and will detect across the bulk of the sample, revealing both surface and sub-surface impurities. If Cr₂O₃ precipitates were within the single crystals, this would be clearly observed in the EPR spectra. To illustrate this and highlight the difference, we have included in Supplementary Information Figure S3 the measured EPR spectra of Cr₂O₃, which shows a distinctly different spectrum with a lower G factor compared to that of lattice incorporated Cr shown in Figure 2a. If Cr₂O₃ like clusters were present in the single crystal grains they would be detected, thus their presence is excluded. To highlight these points, we have included a more detailed description of the EPR results in the revised manuscript. We have also revised the Methods section with respect to the single crystal extraction from the bulk material, emphasising that the material was fractured, liberating the grains, then the crystal grains cut and sonicated to remove adjoined grains and impurity material, particularly that

potentially found in the voids, to obtain a pure product. Such a method is extremely common for crystal separations in general crystal growth science.

2) Related to point 1, it is plausible that under sintering conditions Cr went into solution in the UO₂ lattice but then a significant proportion precipitated out upon cooling. The author should make clear that the results they are showing refer to the end state of doped UO₂ at RT, following potential redistribution of the dopant either between different possible accommodation sites or through precipitation, as well as possible changes in charge state. This should also be discussed in the context of nuclear fuel, which operates at temperatures much greater than that which these experiments were carried out.

Response: We thank the Reviewer for the important comment and have revised the manuscript to emphasise that materials examined are done so at ambient conditions, whereas we discuss the behaviour of Cr at high temperatures, particularly states that occur and its variable behaviour with respect to literature. We agree and acknowledge that at high temperatures it is possible other states of Cr can interact and potentially precipitate out of the UO₂ lattice structure. For instance, in our revised final discussion paragraph (page 23), we have included a discussion of the variable states of Cr found at ambient conditions, but acknowledge the possibility of different states occurring at elevated temperatures such as suggested by Cooper et al.,. We also revised the manuscript to point out that the Cr solubility limit refers to the high temperature state given by Riglet-Martial et al. Here, loss of Cr is possible since we cool under Ar/H₂ atmosphere, which will reduce the oxygen potential, and which thereby is unfavourable for Cr solubility and accordingly is open to loss of Cr via volatilisation. Accordingly, we have included more discussion regarding solubility, behaviour and changes to Cr at HT compared to ambient on P.3 with the introduction and p.21-22 in the discussion. But we also reference the concurrence in comparable result when looking at the works of Mieszczyński et al. who examined fresh and irradiated Cr doped UO₂ in terms of the state of the material when at ambient conditions.

3) In the introduction, the authors mention that dopants improve the power output of the fuel. This is counter intuitive to me as I would expect any dopant to lower the U-loading, and also potentially act as a neutron absorber, therefore reducing the power output. In the same sentence the authors suggest thermal properties are improved - are they referring to thermal conductivity or other properties like specific heat?

Response: We thank the Reviewer for highlighting the limitations of the sentence, indeed we agree with the point a partial loss of power output may arise from the dopant when comparing similar densities of fuels between Cr-doped UO₂ and UO₂. It has been claimed by Westinghouse Nuclear that the increased density of Cr-doped UO₂ compared to standard UO₂ as demonstrated by Kegler et al. (doi.org/10.3390/ma14206160) will result in a net increased U density, further resulting in increased power output. Nevertheless, we agree with the Reviewer that the point is counterintuitive and have refined the sentence to reflect overall that Cr doped fuels present great in-reactor fuel efficiency over UO₂ as demonstrated by previous literature such as by Arborelius, J. et al. Advanced Doped UO₂ Pellets In LWR Applications. Journal of Nuclear Science and Technology 43, 967-976 (2006).

The sentence now reads:

“These include enhanced fission gas retention, increased plasticity, and improved fuel-cladding interactions, thermal properties and leading to overall increased improved in-reactor fuel efficiency over conventional UO₂ in power reactor applications.”

We initially meant by referring to improved thermal properties as a consequence of reduced fission gas release (FGR) that helps improve the loss of thermal conductivity in the fuel cladding gap during

infill with fission gases. Since this improvement is rather a consequence of the improved reduced FGR we have elected to remove it from the sentence to simplify it.

4) The statement "This can be interpreted as a high-spin Cr³⁺ together with oxygen vacancy defects in the vicinity of the substituted Cr site to balance the charge," on page 5 line 104. Do the authors mean that this is the only way the data can be interpreted or is this one of several ways it could be? It would be beneficial if the authors list in a bit more detail other defect complexes and charge states that they considered but ruled out.

Response: We thank the Reviewer for the helpful comment and have provided more discussion on how the EPR spectra can be interpreted and moreover other Cr states be considered/excluded on P.6-7. For example, as highlighted correctly by Reviewer 3/4 it is possible that the two Cr³⁺ cations could be located in the same vicinity with a neutral oxygen vacancy. However, this is discounted due to the detectable lifted anti-ferromagnetic coupling that would result in a considerably different spectra at lower g-value. We provide measured Cr₂O₃ EPR spectra in Supplementary Information Figure S3 to illustrate this point. We also discuss how uncoupled Cr³⁺ and U⁺⁵ would result in a second sharp signal found at lower g-value from the U⁺⁵ centre (Bull. Magn. Reson., 6 (4), 162-224 (1984)). Hyperfine spectra can also result within the EPR spectra but this would arise from electronic coupling to the Cr³⁺ centres. We have discussed these points in on P.6-7 in relation to the EPR spectra we have measured.

5) It is mentioned that Ln³⁺ doping is charge compensated by U⁵⁺, unlike Cr³⁺, which the authors suggest is compensated by oxygen vacancies. There must be a solid physical explanation for this difference? To my mind, Ln³⁺ and Cr³⁺ could, in theory, be charge compensated either way depending on the oxygen potential. The preference would mostly be governed by $V_{\text{O}} + 0.5\text{O}_2 + 2\text{U}^{4+} \rightarrow 2\text{U}^{5+}$, whereby, if the oxygen potential is lowered, V_{O} will become dominant. Differences between the dopants could arise due to binding between the dopant and either V_{O} or U⁵⁺. Do the authors think that the Cr-doped fuel has a lower oxygen potential than for the Ln-UO₂ experiments or do they believe that the dopant- V_{O} and dopant-U⁵⁺ binding energies are dramatically different?

Response: The Reviewer raises a very interesting point regarding the charge compensation mechanism. Based on recent computational work on trivalent metal-ion substituted UO₂, especially the size of the dopant has been shown to play a crucial role in the charge compensation mechanism [e.g. DOI: 10.1016/j.jnucmat.2014.11.089 and 0.1016/j.actamat.2014.06.052].

Generally, two competing effects contribute to the formation energy of solid solutions, endothermic deviation from ideality related to cation size differences and partial stabilization of the system due to the exothermic formation of defect clusters. In redox-active hosts, the redox enthalpy, in this case, the enthalpy of oxidation must be accounted for. Computational work taking into account both charge compensating mechanisms, however, are scarce.

The stability of solid solutions of type U_{1-x}M_xO_{2-0.5x} relative to constituent oxides has been shown to increase with increasing size of the trivalent metal, which explains the general high solubility of e.g. Ln elements in the UO₂ matrix. With increasing cation radius of the trivalent dopant, an increasing preference for higher oxygen coordination has also been established [DOI: 10.1016/j.actamat.2015.06.048]. This has been shown for example for La and Y substituted UO₂, where computational data for doped urania show a decreasing formation enthalpy with increasing dopant radius. In the Y-substituted systems, structures with an enhanced number of vacancies around the Y³⁺ ions are found to be favored energetically, while the opposite is found for La-substituted systems [DOI: 0.1016/j.actamat.2014.06.052]. The different charge compensation mechanisms for larger and smaller lanthanide cations also seem to depend on the cation size of the dopant. For La-substituted urania, charge compensation via the oxidation of U⁺⁴ to U⁺⁵ is well-established [e.g. DOI:

10.1039/C6CC07684J and 10.1021/acs.inorgchem.7b02839]. For the smaller lanthanide Gd, substitution in urania results in the formation of oxygen vacancies for charge compensation [DOI: 10.1149/2.047403jes and 10.1016/j.electacta.2017.07.006] .

As the energetics of solid solution formation and associated charge compensation mechanisms are very complex and warrants detailed studies, we can only rationalize the charge compensation mechanism in the Cr⁺³-doped UO₂ structure considering the differences between dopant and host cation radii. Cr⁺³ with a coordination number of six, has a cation radius of 61.5 pm. Uranium in oxidation states +IV and +V are significantly larger, with radii of 86 and 79 pm, respectively, for the same coordination.

In solids where the aliovalent dopant is smaller than the host cation (such as Cr⁺³ in UO₂), the dopant shows a tendency to attract oxygen vacancies, thereby reducing its coordination number. Moreover, the introduction of charge compensating oxygen vacancies, which are slightly larger than the oxygen anions in the lattice, partly compensate for the lattice contraction observed for small dopants [DOI: 10.1016/0022-3115(81)90544-4]. If on the other hand oxidation of U⁺⁴ to U⁺⁵ for charge compensation is considered, the redox reaction would result in further contraction of the urania lattice, and can thereby be considered unfavorable, with the increased lattice energy deviation from ideality.

The formation of Cr⁺³ and oxygen vacancies is further supported by our EPR studies and U⁺⁵ is not supported since it is not observable in the EPR spectrum.

6) I would like the authors to elaborate more on why they believe the proportion of Cr in the 2+ state is so certainly in a second phase, given that CrO does not appear on Cr-O phase diagram at these temperatures. As far as I am aware, CrO forms as a liquid at sintering temperatures but should be converted to Cr metal and Cr₂O₃ when cooled. I understand the argument that the lack of Cr²⁺ in the single crystal data indicates that it is not in the bulk lattice but, if the CrO precipitate argument is invoked, it should be explained how this does not contradict the phase diagram. Could Cr²⁺ be incorporate as individual atoms within the grain boundary and then get chipped off during mechanical processing?

Response: This is a very pertinent and helpful point made by the Reviewer and we thank them for highlighting this. We have considered the possibility of CrO in our samples based on the phase diagram of Cr-O and also recognising the slow cooled conditions used in our sample preparation which is now listed in the manuscript and methods (6 °C/min). Subsequently we have ruled it out as now discussed on P.21-22 as quenched CrO would not occur in these conditions. Although from our EXAFS analysis the Cr⁺² appears to have some "CrO" identity, we agree with relevant literature by Riglet-Martial et al. and Leenaers et al. that this can result from the incomplete decomposition of liquid CrO with cooling resulting in substoichiometric Cr₂O₃ states in which they observe also. The possibility of Cr⁺² existing within Cr₂O₃ when substoichiometric has also been previously highlighted in literature (Biesinger et al. Cabrera et al. and Payn et. Al). Very recently Devillaire et al. proposed the occurrence of metastable Cr₃O₄ additionally in these regions via Raman spectroscopy (<https://doi.org/10.1002/jrs.6512>). We suspect during the crystal separation process when grain boundary Cr₂O₃-Cr-"CrO" material is liberated from the crystal grains it rapidly sees atmosphere and oxidises. That the Cr⁺² appears CrO like from the EXAFS is attributed to the local environment being enriched in Cr-O material. A recent study by Middelborough et al. highlights this extensive enrichment. We provide a long discussion regarding these points on chemical identity of Cr⁺² on p. 22-23.

7) To what accuracy are the authors certain their undoped UO₂ sample is exactly stoichiometric? Figure 3 is used to show that additions of Cr do not increase the U⁵⁺ concentrations, thereby

indicating Cr is not charge compensated by U⁵⁺. However, if there is even a very small amount of non-stoichiometry in UO_{2+x}, approximately x=0.001, the U⁵⁺ concentration would be governed by the concentration of the oxygen interstitials, rather than by the potentially even smaller quantity of Cr in solution. In this case, Cr could still be charge compensated by U⁵⁺ but it would have a negligible impact on the overall U⁵⁺ concentration. I did these estimates pretty roughly but the authors should demonstrate that this possibility does not affect their conclusions.

Response: This is another important point which we considered in our experimental plan and have now provide the Ellingham diagram in Supplementary Information Section 6 (shown below also for reference) showing conditions used for the experiment w.r.t UO₂ expected stoichiometry. Included now in the methods, we describe that the samples were cooled under gas flow atmosphere slowly, thermodynamically this will cause the oxygen partial pressure to decrease with temperature becoming more reducing. From the Ellingham diagram a stoichiometry of UO_{2.001} is accordingly not expected to be encountered. Following the trajectory of the sample from the diagram shows it will largely track the UO_{2.0} line and head towards the UO_{2.0001} line as an extreme. If the stoichiometry is calculated assuming this maximum limit of UO_{2.0001} whilst assuming 750 ppm Cr⁺³ doping in the lattice, a stoichiometry of Cr_{0.00388}U_{0.99612}O_{1.99816} will be obtained (converting from weight ppm to Mol). The Cr⁺³ excess in comparison to U⁺⁵, which is created to compensate for the hyperstoichiometry in UO_{2.0001} is almost 20-fold. In other words, the presence of U⁺⁵ is not sufficient to compensate the charge mismatch resulting from Cr⁺³ incorporation. A similar conclusion can be made from calculation for both UO_{2.001} and UO_{2.0005}. **We have included an excel file for the Reviewers detailing the calculations title “Cr calculations for revision”.** Accordingly, another charge compensation mechanism via the introduction of defects will be required even at this maximum assumption brought on by the inclusion of Cr into the lattice. Indeed, it is difficult to properly quantify these numbers, however pertinently EPR has the resolution to detect U⁺⁵ forming from sample hyperstoichiometry, if a stoichiometry of UO_{2.001} were encountered, for instance via problems in maintaining atmosphere in the furnace a sharp signal at low g-factor would be detected (“EPR of Uranium Ions”, I. Ursu and V. Lupei Bull. Magn. Reson., 6 (4), 162-224 (1984).]. We have revised the manuscript to reflect the strength of EPR detection of low concentrations of U⁺⁵.

8) One very minor comment: figure 2 seems to be a jpeg - it would look better as png or pdf.

Response: We thank the Reviewer for the comment and have both re-formatted and re-exported Figure 2.

Reviewer #2 (Remarks to the Author):

The work by Murphy et al. seeks to resolve the discrepancy in the literature regarding the oxidation state of Cr in UO₂. The topic is of importance for the nuclear community and the use of EPR on this class of materials is novel to the best of the Reviewer's knowledge. Though, the more important information is how the Cr facilitates improved properties during sintering, which would require in situ studies rather than measurements under ambient conditions. The work may be better suited to a nuclear materials focused journal. That said, the methodology and conclusions drawn are overall logical and appropriate. Below are a few comments and questions which should be addressed/considered.

1) Page 2: line 46, 47, in fact, the solubility of Cr into UO₂ is subject to debate. Some claim a solubility limit as high as double what the authors claim here [Cardinaels et al. Journal of Nuclear Materials 424. (2012) 252-260]

Response: We thank the Reviewer for the comment and agree that the solubility limit is extremely sensitive to conditions used during material preparation. It has been established by the works of Riglet-Martial et al. (<http://dx.doi.org/10.1016/j.jnucmat.2013.12.021>) and Leenaers et al. (doi:10.1016/S0022-3115(02)01693-8) that typically the solubility of Cr into UO₂ is dependent upon the temperature and oxygen partial pressure, whereby higher temperatures and high oxygen partial pressures increase solubility as demonstrated by Riglet-Martial et al. Recently Kegler et al. (<https://doi.org/10.3390/ma14206160>) showed, it can also be effected by the method used for synthesis (dry mixing, wet coating, coprecipitation). Furthermore, loss of Cr via volatilisation as demonstrated by Peres et al. (doi:10.1016/j.jnucmat.2012.01.001) which can be above 30 % also effect solubility. Another factor effecting the solubility is cooling method, which, when under continuous gas mixed atmosphere results in reduced oxygen partial pressure which result in loss Cr as well. In the case of Cardinaels et al., their investigation used higher temperatures in their material preparation than the current. We used conditions of 1700 °C and -420 kJ/mol, which by Riglet-Martial et al. results in a solubility of 750 ppm. This value carries an uncertain of approximately ± 50 kJ/mol via the experimental conditions and aforementioned factors. Inspecting Figure 2 by Cardinaels et al. shows this is consistent with their work. Moreover, we have significantly increased the discussion in the manuscript in both the Introduction and Discussion sections regarding the variably solubility of Cr into UO₂ based on synthesis conditions see P. 3 and 22-23.

2) Page 3 line 74 No argument that the extracted single crystals were completely representative of the material overall, but what's the basis for this claim. How many extractions performed and how do they compare to one another and the bulk material structural and electron spectroscopic measurements? It appears only one single crystal diffraction pattern was analyzed, and the unit cell parameters are quite different than that of the bulk.

Response: We thank the Reviewer for the good comment. In the course of the work, we extracted several single crystal grains. For ones that were subject to ultimately EPR, HERFD-XANES and EXAFS, hemisphere data collections were undertaken for each individually. A significant number of crystals were extracted particularly for EPR, due to its sensitivity in measurement and subsequent want to achieve suitable quality measurement. We have now included in the manuscript the single crystal X-ray diffraction refinements for SC-2 in the SI Table S3. **In addition, 4 additional refinements made against other extracted single crystal grains have been submitted as a separate file ("Single_Crystal_Additional_Refinements") for the Reviewers.** Comparing the 6 refinements shows almost identical values in lattice parameters. Comparing the average lattice volume of the 6 crystal refinements to the synchrotron X-ray diffraction refinement of the bulk material they're extracted from, gives a difference of approximately 0.6 %. Considering that the single crystal X-ray diffraction measurements uses a different source, geometry, set of optics and is ultimately a different

measurement technique, we believe the offset is acceptable. We also note a significant discussion in literature has been made regarding the accuracy of lattice parameters between single crystal and powder diffractometers (F. H. Herbst *How precise are measurements of unit-cell -dimensions from single Crystals?* <https://doi.org/10.1107/S010876810000269X>). Herbst describes that systematic errors between the measurement types can be caused often using area detectors which the Agilent Oxford Diffraction Super Nova single crystal diffractometer used in the present study possesses.

3) Page 6 line 127 what is the energy resolution of the HERFD-XANES measurements performed here?

Response: For Cr, a combined (incident convoluted with emitted) energy resolution of 1.1 eV was obtained as determined by measuring the full width at half maximum (FWHM) of the elastic peak. We have now included these details in the methods section.

4) Page 9 line 177 "... compared to the single crystal grain The spectrum..." missing period between sentences.

Response: We thank the Reviewer for identifying the error, have corrected it.

5) Page 9 line 190 "Figure (c) and (d)" needs figure number as well. I presume figure 2c and 2d.

Response: We thank the Reviewer again for the identifying the error, corrected it.

6) Which single crystal was diffraction performed on (SC1 or SC2)? If both were measured, how do they compare to one another and to the bulk powder?

Response: The presented single crystal diffraction data in the original manuscript (Supplementary Information Table S2) was for SC1. With comment 2) we have now included the data for SC2 for comparison and also provide data for 4 other crystals extracted in the course of our work. Comparing the 6 crystals shows precise results although less accuracy with an average difference of 0.6 % in unit cell volume than the synchrotron powder diffraction results as previously discussed. Compared to SC1, SC2 has larger R values.

7) Valid that the EPR data shows no evidence of U⁵⁺, but not convinced that HERFD-XANES possesses the sensitivity to corroborate these results. The signal is dominated by U⁴⁺ and lacks resolution to observe a small shift associated with ppm levels of U⁵⁺.

Response: We thank the Reviewer for the comment and agree with the point. Indeed the EPR provides far superior resolution over the HERFD-XANES in detection of ppm levels of U⁵⁺. Accordingly, we have opted to move the HERFD-XANES results for U on the M and L-edge to the Supplementary Information and provide further discussion/clarification in the text, that the EPR gives greater measurement control for U⁵⁺ detection whereas the U M and L-edge show overall comparability in the U redox states between the sample types (pure UO₂ powder, Cr-doped UO₂ powder and Cr-doped UO₂ single crystals). We also discuss the possibility and lack thereof for hyperstoichiometry and subsequent U⁵⁺ in response to the pertinent question from Reviewer 1 Question 7 based on the Ellingham diagram of UO₂.

Reviewer #3 and #4 (Remarks to the Author):

The paper addresses the oxydo-reducing properties of chromium within doped UO₂ nuclear fuel. It aims at identifying the valence state(s) of chromium within the material, in particular in the fluorite-type UO₂ phase, using advanced spectroscopic techniques.

Assessment of chromium oxide as dopant for UO₂ system is nowadays more than ever of great current interest for regulator and fuel vendor. Much research is in progress and still needed to clarify the role of additives in UO₂ fuel as regards the material properties (gas transport, precipitation or release of fission gas, mechanical behaviour, thermal conductivity, etc.). The aim is to design nuclear fuel material showing safe behaviour (Accident Tolerant Fuels) as well as optimised performances (limitation of fission gas release, prevention of clad failure by I-CSC, etc.) in both normal and off-normal operating conditions.

As such, the topic addressed in this article is of prime current interest and worthy for publication in Nature Communications.

It is well known, from a number of papers available in the open literature, that chromium oxide is very sensitive to oxido-reduction in the conditions relevant for fuel fabrication and nuclear LWR fuel operation.

As such :

*- the speciation and solubility of chromium in UO₂ is very much dependant on both temperature and oxygen partial pressure, which may result in a great variety of UO₂ fuel microstructures;
- the behaviour of Cr-doped fuel in operation (normal and off-normal conditions) is essentially determined by the oxido-reducing conditions (pO₂ versus T) taking place within the fuel pellets. The complexity of the chemical speciation of chromium in conditions relevant for nuclear fuels as well as its high sensitivity to redox variations require great experimental caution in the dedicated experimental work.
In the present study, the scientific work is well conducted, with great experimental care, and give rise to high quality data. In contrast to previous investigations, special attention is paid to physically separate the Cr-doped UO₂ matrix crystals from the chromium precipitates in separate phase, in order to overcome the difficult problem of convolution of the spectroscopic signals and to identify unambiguously the chemical states of chromium within the material.*

Some minor points to consider in the final version:

Response: we would firstly like to thank the Reviews profusely, for the extremely productive and thorough reviews which have helped improve the manuscript greatly, and pointed out factors which were overlooked.

1.) As solubility of chromium in UO₂ varies as a function of temperature and oxygen potential, it is essential to systematically explain the (T, μ O₂) conditions related to the specified solubility values. In addition, it is also essential to specify whether the given solubility values refer to ppm Cr in UO₂ or ppm Cr₂O₃ in UO₂ (which is quite different).

Response: We thank the Reviewers for the helpful comment and advice, we have modified several parts of the manuscript to firstly discuss the importance of synthesis conditions (particularly temperature and oxygen potential) on the solubility of Cr into UO₂ (P.2-3, 20-21) whilst also referring the conditions used for material preparation. Indeed, we have also clarified several points of the material preparation including specific conditions (temperatures, oxygen partial pressure, hold time

and ramp rates) and also the Cr addition, which as correctly identified by the Reviewers is as ppm of Cr_2O_3 .

2.) In the preparation section, the amount of chromium added to the samples need to be more clearly defined : Is it "addition of 3500 ppm Cr" or "addition of Cr corresponding to 3500 ppm Cr_2O_3 in UO_2 " in consistency with Kugler et al. previous study ?

Response: We thank the Reviewers for highlighting the important point in our materials preparation, and as correctly identified we used ppm of Cr_2O_3 consistent with Kegler et al. previously.

3.) The word "purity" is used in order to state that the single crystal is devoid of secondary chromium bearing phases, but it may lead to confusion with regards to the chemical purity of UO_2 itself (without any ppm of Ca or Fe...). Clarification needed.

Response: We thank the Reviewers for the helpful comment and have amended the manuscript accordingly, removing the term "high purity" in the abstract and other sections where clarification is missing using the suggested phrases by the Reviewers.

4.) The recent publication <https://doi.org/10.1038/s42004-022-00784-3> (December, 1st, 2022) could be additionally discussed in order to explain whether the observed differences in the oxidation state of chromium come from irrelevant thermodynamic conditions (uncontrolled O_2 partial pressure, more reducing conditions?) or from secondary phases bearing chromium (quenched liquid CrO ?) which were too much represented in the sample.

Response: We are aware of this recent publication, which has assigned the Cr^{+2} state to the lattice environment in calcined UO_2 bulk material. Exact details of the synthesis conditions, such as the oxygen partial pressure, are not given in the paper. However, by inspecting their SEM micrographs of their sintered material, which show a clearly smaller size of the UO_2 grains in comparison to the material in our study, implies that a lower oxygen partial pressure was used. This would further mean that less CrO(L) and Cr_2O_3 were formed at high temperature and that their sample thereby has a significant amount of Cr metallic (deduced from the phase diagram of Cr and considering CrO(L) is needed to enhance grain growth). We believe that the metallic Cr signal can be seen in their XANES spectra, particularly when comparing to the HERFD-XANES Cr^0 reference spectrum given in our manuscript. This is consistent with the Reviewer's point that likely they used conditions, which over-represent reduced Cr states in the bulk material. As shown in our manuscript, the presence of multiple Cr oxidation states in the bulk material renders the definite assignment of the Cr oxidation state in the UO_2 lattice difficult. A drawback of the recently published paper is the indirect assignment of Cr^{+2} in the lattice, as only BVS and EXAFS have been used to identify the presence of Cr^{+2} . Although EXAFS is a powerful method for the determination of the Cr coordination geometry in the samples, a precise assignment of the oxidation state would require (high-resolution)-XANES investigations and comparisons to Cr oxidation state standards for all plausible oxidation states (based for example on the Cr speciation obtained in the Ellingham diagram) as we have used in our study. The BVS method is a known approximation of the valence, but not a direct measurement of oxidation state. In our manuscript we use EPR and HERFD-XANES which are sensitive and well-established methods for oxidation state determination. Finally, the authors have identified rather substantial concentrations of U^{+5} in the calcined UO_2 material, up to 32 wt%. The presence of U^{+5} has been suggested in that study to balance the charge of the subvalent Cr^{+2} cation in the UO_2 lattice. With this large concentration of U^{+5} , it is indeed likely that no additional charge-compensating mechanisms are required to accommodate the subvalent Cr cation. However, we would not expect a visible increase of the U^{+5} concentration in the sample with the incorporation of the minute amount of Cr^{+2} (100-2400 ppm) (we also demonstrate this via the pertinent question from Reviewer 1). As a significant part of the discussion in their manuscript is based on the pre-calcined UO_2 material, the experimental conditions

do not compare to the ones that we have used in the current manuscript, and by extension to nuclear fuel pre- and post-reactor irradiated material, which we target and is of most relevance. We have thereby cited this recent study on p.22 and p.23 in the revised manuscript, however, further discussion and comparison between that study and ours have not been made. Moreover, we have provided a detailed discussion on p.22-23 regarding the different Cr states that can be found within the bulk material, which can convolute measurement and further are dependent on the synthesis conditions. We hope that it is thereby clear to the readers, that discrepancies in the assignment of Cr oxidation states in previously published studies may be related to the problems of measuring bulk material, which can over-represent the amount of reduced Cr states in the material.

5.) In figure 5, the only possibility given is a Cr³⁺ with a neighbouring oxygen vacancy with charge +1. Couldn't it be two neighbouring Cr³⁺ with a neutral oxygen vacancy? Would this latter case lead to a hyperfine structure in Cr ³⁺ EPR spectrum?

Response: This is a good point raised by the reviewers. If two Cr³⁺ cations were located in the same vicinity this would result in a detectable spectra at lower g-value comparable to the now included EPR (Supplementary Information Section 4) spectrum of Cr₂O₃ which becomes observable due to the lifting of the anti-ferromagnetic coupling (DOI: 10.1080/00337577008234983). Accordingly, we can rule this mechanism out as occurring in Cr-doped UO₂ since we do not observe it in our EPR spectra. Hyperfine spectra could occur due to electronic coupling to the lone Cr³⁺ centres, but we do not observe this in our measurements.

6.) The consistency between the present spectroscopic observations and thermodynamics should be discussed more extensively.

Response: We take note of the Reviewers excellent comments regarding the consistency between the HERFD-XANES results of the bulk material and the thermodynamic calculations. In our material synthesis, we used 3500 ppm as Cr₂O₃, which corresponds to 2395 ppm as Cr. If we assume a 30 % loss of volatilisation of Cr, in agreement with the work of Peres et al. (doi:10.1016/j.jnucmat.2012.01.001), this would result in approximately 1680 ppm of Cr remaining in the material. Our HERFD-XANES ITFA decomposition yields 65% Cr³⁺, 26 % Cr²⁺, and 9% Cr metallic. However, the ITFA analysis cannot distinguish between (Cr⁺³_xU⁺⁴_{1-x})O_{2-0.5x} and Cr⁺³₂O₃. Thereby, we have to base the amount of incorporated Cr³⁺ on the known solubility of Cr³⁺ in UO₂ at the synthesis conditions applied in this study, i.e. a solubility of 750 ppm at 1700° C and -420 kJ/mol. Accordingly, the Cr concentrations would be: 750 ppm (Cr⁺³_xU⁺⁴_{1-x})O_{2-0.5x}, 340 ppm Cr₂O₃, 436 ppm Cr²⁺, and 151 ppm Cr⁰. These numbers are indeed well in alignment with the thermodynamic studies of Riglet-Martial et al. among others. We note the calculations made by the Reviewers, we believe they may have made an error in the conversion from Cr₂O₃ ppm to Cr ppm. Importantly we acknowledge that there is a considerable error in our calculations, firstly in the HERFD-XANES which carries a 5 % uncertainty, the 30 % volatilisation by Peres et al. is based on 4 hours at temperature maximum, whereas we used 10 hours. We also cooled under gas atmosphere, which would decrease the oxygen partial pressure, meaning Cr maybe lost from the lattice. We have discussed these calculations on p.12 but also emphasised the potential uncertainty in addition to more over agreeing and highlighting the good consistency between the experiments and previous thermodynamic predictions. We also make further references to the agreement in the discussion and the conclusion.

7.) Two references requires completion.

Response: we thank the Reviewers for highlighting the errors in the references and also referencing structure, we have corrected them.

8.) Minor typographic inaccuracies require correction.

Response: we thank the Reviewers for highlighting typographical errors throughout the manuscript, we have corrected these thanks to the help of the Reviewers whilst carefully checking for other errors.

Extended comments are included in the enclosed files.

Response: we thank the Reviewers for the extended comments and review in the supplied files. We have responded to all the points raised in the file, where when relevant referred back to this Reviewer file for conciseness.

REVIEWERS' COMMENTS

Reviewer #1 (Remarks to the Author):

I thank the authors for their comprehensive response to my comments and I am happy to recommend publication.

Reviewer #2 (Remarks to the Author):

The authors have appropriately addressed all comments and suggestions. I recommend this work to be published.

Reviewer #3 (Remarks to the Author):

The authors of the paper made all the necessary changes to their work, considering the first round of comments.

Upon minor corrections included in the enclosed file, the present revised manuscript is suitable for publication in Nature Communications.

Reviewer #4 (Remarks to the Author):

Dear Authors,

I thank you for this new release. As far as I am concerned the manuscript can be accepted with minor remarks or changes (it is up to you to decide). My comments appear with the label "JL118404" in the pdf file.

My main remarks are that :

- equilibrium of the chromium saturated UO₂ matrix with liquid CrO phase leads to an exaggerated grain growth but grain growth also occurs to a lesser extent when chromium saturated UO₂ matrix is in equilibrium with Cr₂O₃ (but not when it is in equilibrium with metallic Cr). So that occurrence of liquid CrO should not be presented as the only grain growing process,
- since no twin Cr³⁺ exist sharing an oxygen vacancy, is the electrical balance realized when Cr⁺³ replaces U⁺⁴ with an electron in the oxygen vacancy ?

Best regards

Reviewer Response

Reviewer #1 (Remarks to the Author):

I thank the authors for their comprehensive response to my comments and I am happy to recommend publication.

Response: We thank the Reviewer for the extremely thorough review of our manuscript which has assisted in adding pertinent technical and scientific points to the manuscript. We are also highly appreciative of the recommendation for publication.

Reviewer #2 (Remarks to the Author):

The authors have appropriately addressed all comments and suggestions. I recommend this work to be published.

Response: We thank the Reviewer for the productive and beneficial review that has greatly improved the condition of the manuscript. We are grateful for the recommendation for publication.

Reviewer #3 (Remarks to the Author):

The authors of the paper made all the necessary changes to their work, considering the first round of comments.

Upon minor corrections included in the enclosed file, the present revised manuscript is suitable for publication in Nature Communications.

Response: We thank the Reviewer again for the overall highly constructive review from both rounds of revision. We have responded to each point made in the annotated PDF in the last section of this document. We are also highly appreciative and thankful for positive the recommendation for publication following hopefully satisfactory checking of the final points.

Reviewer #4 (Remarks to the Author):

Dear Authors,

I thank you for this new release. As far as I am concerned the manuscript can be accepted with minor remarks or changes (it is up to you to decide). My comments appear with the label "JL118404" in the pdf file.

My main remarks are that :- equilibrium of the chromium saturated UO₂ matrix with liquid CrO phase leads to an exaggerated grain growth but grain growth also occurs to a lesser extent when chromium saturated UO₂ matrix is in equilibrium with Cr₂O₃ (but not when it is in equilibrium with metallic Cr). So that occurrence of liquid CrO should not be presented as the only grain growing process,

- since no twin Cr³⁺ exist sharing an oxygen vacancy, is the electrical balance realized when Cr³⁺ replaces U⁴⁺ with an electron in the oxygen vacancy ?

Best regards

Response: We thank the Reviewer profusely for positive recommendation and overall highly helpful reviewer that has benefited the manuscript and content superbly. We agree with the Reviewers point regarding the importance of Cr chemical factors, not just CrO, in driving and contributing to grain growth in Cr-doped UO₂. We have adjusted the text to reflect this, since without, it would imply CrO is the only factor, which the Reviewer correctly points out, is not true and can be assisted by Cr₂O₃ etc.

Since no twin Cr³⁺ exist sharing an oxygen vacancy, is the electrical balance realized when Cr³⁺ replaces U⁴⁺ with an electron in the oxygen vacancy

Response: This is an important observation made by the Reviewer which we have consulted at length. Different types of oxygen vacancies (O_{vac}), such as +2 charged (V_O^{**}), +1 charged (V_O^*) and neutral (V_O^x) defects, can form within a doped lattice, depending on intrinsic defects in the material and the valency of the dopants. The paramagnetic defect, i.e. V_O^* should be readily observable by EPR, producing a sharp signal in the EPR spectrum at $\sim g = 2$, while a +2 or neutral vacancy is EPR silent. In the current study, we do not observe any signal of paramagnetic vacancies in our EPR spectrum, which implies

that the generated oxygen vacancies following Cr³⁺ substitution in the UO₂ lattice must be +2 or neutrally charged on average.

This is in agreement with the common way of describing the formation of oxygen vacancies for trivalent dopants in tetravalent hosts:

For every two subvalent dopants substituting for two host cations, one oxygen vacancy with a charge of +2 is created in the lattice. The coordination of the doubly charged vacancies to the subvalent cation can form dimers M'–V₀^{••} or trimers M'–V₀^{••}–M' as shown for Ce³⁺ doped CeO₂ at low oxygen potential (<https://doi.org/10.1103/PhysRevB.87.134104>). Middleburgh et al. calculated the defect formation mechanisms for Cr₂O₃ doped into UO₂, finding that Cr⁺³ centres are isolated, consistent with our EPR results, and involve vacancy formation of the type 1V₀^{••} as that of CeO₂ (<https://doi.org/10.1016/j.jnucmat.2011.10.006>).

We agree with the Reviewer that the occurrence of a ½ defect with two electrons, although charge balances, is not physically meaningful. Vacancies, however, are highly mobile and they can recombine (as an example, singly charged vacancies can be reduced to neutral vacancies by trapping an additional electron). This way, the charge in the lattice is globally balanced, although local charge imbalances may exist. Indeed, it has been shown in the case of UO₂ (Pavlov et al. <https://doi.org/10.1016/j.actamat.2017.07.060>) that at temperatures similar to those used in the present studies sintering of Cr-doped UO₂, oxygen defect interactions become significant and thermally activated hopping of defects (polarons) is dominant. This includes the occurrence of defect recombination and exchange effects.

In the current study, we show the included Cr⁺³ centres do not interact from EPR in fresh sintered Cr-doped UO₂, and this is also consistent from post reactor irradiated Cr-doped UO₂ in the case of Mieszczynski et al. by comparing XANES spectra, and further the inclusion can only be achieved via oxygen defects and not U⁺⁵. However, it is difficult to assign the exact nature of the oxygen defects in the UO₂ crystal lattice induced by Cr⁺³ between initial-instantaneous incorporation and post sintering from the current measurements, since we measure the end result.

As we do not have precise knowledge of the generated vacancies in the UO₂ lattice, but we can exclude paramagnetic defects, we have revised Figure 4 to include an oxygen vacancy (V₀) in the Cr³⁺ coordination shell without specifying its charge.

Response to Annotated PDF comments

I would keep the O of Cr⁺² O(liq.) just as Cr₂O₃ is written in the form Cr⁺³ 2O₃, maybe would it be useful to add the fact that it is a liquid phase since it is less known than other chromium oxides.

Response: We thank the Reviewer for the comment, although we do agree with the Reviewer that the Cr⁺² is ultimately arising from the former Cr⁺²O(l) state, it was highlighted by Reviewer 1 in the previous revision that having it is as Cr⁺²O is counter intuitive since it is not stable at ambient conditions and it would imply it's been obtained via quenching which via the slow cooling used isn't possible. We further note the very recent tentative observation of Cr₃O₄ using Raman spectroscopy by Devillaire et al. in Cr doped UO₂ (<https://doi.org/10.1002/jrs.6512>). This phase is balanced containing Cr⁺² when compared to the Co₃O₄ relation. Accordingly, although Cr⁺² is found in substoichiometric Cr₂O₃ as we discuss on P.21, this Cr₃O₄ phase could also be a dual source. Nevertheless, consistent for both is that the Cr⁺² is associated with the oxide precipitate and grain boundary regions i.e. non-lattice, accordingly we have adjusted the sentence to reflect this.

I would say either "the understanding of discrepancies" or "understanding discrepancies"

Response: We thank the Reviewer for the helpful comment and have opted for the former.

(pO₂) or oxygen potential ($\mu_{O_2} = RT \ln(pO_2)$)

Response: We thank the Reviewer for identifying the error and have corrected the text to be potential and not pressure since we quote oxygen potentials

Liquid CrO is part of eutectic compositions. These eutectic compositions are mainly liquid CrO with small amounts of U dissolved in it.

So that in the sentence, I would set "liquid CrO" just after "eutectic compositions"

that is :

"eutectic compositions (liquid CrO), metallic Cr and..."

Response: We thank the Reviewer for the helpful comment and have adjusted manuscript accordingly.

In fact, saturation of UO₂ with Cr³⁺, either by means of the equilibrium with Cr₂O₃ or CrO, leads to grain growth. But grain growth is exaggerated by means of an equilibrium with liquid CrO because of the dissolution precipitation of U in liquid CrO

Therefore, I would rather say:

"Indeed, the existence of dissolved Cr³⁺ in UO₂ by means of its equilibrium either with Cr₂O₃ or liquid CrO..."

Response: We thank the Reviewer again for the extremely helpful comment, we have used the sentence they have recommended but rather than saying "dissolved Cr³⁺ in UO₂" we use "Cr", since this section is a part of the introduction and the manuscripts' purpose is to demonstrate it is indeed Cr³⁺ rather than assume it in this point of the manuscript.

I would say "emphasized"

Response: We thank the Reviewer for good suggestion and have used it.

Despite

Response: We thank the Reviewer for good suggestion and have used it.

Does it mean that the oxygen vacancy contains an electron ? In that case is it VO[•] instead of VO^{••} ? What would be the EPR signal in that case ?

Response: This is a very pertinent question by the Reviewer, paraphrasing our response to Reviewer 4 described earlier: Static structurally, the isolated Cr³⁺ centres which have been conclusively identified by the EPR and found not be interacting with each other, substitute for U⁴⁺, where for perfect substitution would involve an oxygen defect forming in which for charge balancing contains 1 electron. However, such a species should be readily observable by EPR producing a sharp signal in the EPR spectra at $\sim g = 2$ in which it is not observed in our spectrum. We agree with the Reviewer that the occurrence of a $\frac{1}{2}$ defect with two electrons, although charge balances, is not physically meaningful. Vacancies, however, are highly mobile and they can recombine (as an example, singly charged vacancies can be reduced to neutral vacancies by trapping an additional electron). This way, the charge in the lattice is globally balanced, although local charge imbalances may exist. Indeed, it has been shown in the case of UO₂ (Pavlov et al. <https://doi.org/10.1016/j.actamat.2017.07.060>) that at temperatures similar to those used in the present studies sintering of Cr-doped UO₂, oxygen defect interactions become significant and thermally activated hopping of defects (polarons) is dominant. This includes the occurrence of defect recombination and exchange effects.

In the current study, we show the included Cr³⁺ centres do not interact from EPR in fresh sintered Cr-doped UO₂, and this is also consistent from post reactor irradiated Cr-doped UO₂ in the case of Mieszczynski et al. by comparing XANES spectra, and further the inclusion can only be achieved via oxygen defects and not U⁵⁺. However, it is difficult to assign the exact nature of the oxygen defects in the UO₂ crystal lattice induced by Cr³⁺ between initial-instantaneous incorporation and post sintering

from the current measurements, since we measure the end result. As we do not have precise knowledge of the generated vacancies in the UO_2 lattice, but we can exclude paramagnetic defects, we have revised Figure 4 to include an oxygen vacancy (V_O) in the Cr^{3+} coordination shell without specifying its charge.

:

Response: Corrected.

Be careful, ITFA is only defined later on at line 236

Response: Good observation, will depend on the final journal edited version of the manuscript but we have stated the acronym in any case.

I suggest deleting this sentence which is redundant with the next one. Additions in the analysis of the data are proposed in the last sentence of the paragraph.

Response: Although we agree with the Reviewer, we have opted to leave the first sentences in the text. The text is written for a more general scientific audience, such that although this discussion is perhaps more redundant to those more familiar with the topic, we feel it is more readily understood by a general scientific audience with the extra detail and better appreciated regarding the overlap between the present results and previous thermodynamic estimates.

However, under

Response: corrected

left in UO_2

Response: Have opted for "in the bulk material"

(65%)

Response: corrected

between

Response: corrected

Considering

Response: corrected

Given the experimental and analytical uncertainties, the proportion of chromium in the Cr₂+2O₃ phase (20%) would be about double that in the CrO phase (9%), which is fully consistent with the possible uncomplete disproportionation of the unstable Cr₂+O phase upon slow cooling according to the main reaction $3\text{Cr}_2+\text{O} = \text{CrO} + \text{Cr}_3+2\text{O}_3$, as suggested in reference 15.

Response: This is an excellent observation by the Reviewer, we have included this in the immediate text but also in the discussion regarding the distribution of phases and origin. We thank the Reviewer for the great observation.

This interpretation is perfectly in line with the known Cr-O phase diagram of Toker (reference 22) and supports

Response: Very good observation, the sentence has been amended.

as well as

Response: Sentence has been adapted.

Unclear

Response: Have opted to delete the section of the sentence, it does not add anything more than what is present we have decided.

maybe "those" instead of "what"

Response: agree, have changed the sentence.

Electrically $1/2\text{VO}^\circ$ balances the charge due to the substitution of Cr⁺³ by U⁺⁴. But either 1 oxygen is missing or not, i.e. either there is 1 vacancy or not. There cannot be half vacancies in the structure.

Does it mean that the number of oxygen vacancy per Cr⁺³ is 1 and that the vacancy contains 1 electron? It is the only possibility I can imagine since oxygen vacancies are not shared by neighbouring Cr⁺³ ions.

Response: This is an excellent question by the Reviewer, we have included an extended response for this on the previous page to Reviewer 4, discussing this and also the limitations from experimental measurement.

In order to balance electrical charges (substitution of U⁺⁴ by Cr⁺³), there should be an electron in the oxygen vacancy so that it should be VO[°] instead of VO^{°°}. Or is it not the case

Response: Again, this is an excellent question by the Reviewer, we have included an extended response for this on page 4 of this document, discussing this and also the limitations from experimental measurement.

scattering ?

Response: well spotted error, corrected.

Cr+2

Response: corrected

affected ?

Response: corrected

I just wanted to add that although CrO liquid leads to exaggerated grain growth, Cr₂O₃ also leads to enhanced grain growth to a lesser extent. Grains are larger when Cr₂O₃ is coexists with chromium-saturated UO₂ than without any dopant

Response: We thank the Reviewer for the very helpful comment, we have adjusted the sentence to reflect the Cr₂O₃ also plays a role in the grain growth process.

According to

Response: corrected

Who

Response: corrected

The present

Response: corrected

:

Response: corrected

proportions

Response: corrected

Are

Response: corrected

Vol.54, Issue 2 (February 2023)

Response: corrected – the original citation was made before the associated journal provided the fuel details